# Human TRPC5 structures reveal interaction of a xanthine-based TRPC1/4/5 inhibitor with a conserved lipid binding site

David J. Wright [ID] [1,2], Katie J. Simmons[1,2], Rachel M. Johnson[2,3], David J. Beech [ID] [1], Stephen P. Muench [ID] [2,3 ✉] & Robin S. Bon [ID] [1,2 ✉]

TRPC1/4/5 channels are non-specific cation channels implicated in a wide variety of diseases, and TRPC1/4/5 inhibitors have recently entered clinical trials. However, fundamental and translational studies require a better understanding of TRPC1/4/5 channel regulation by endogenous and exogenous factors. Although several potent and selective TRPC1/4/5 modulators have been reported, the paucity of mechanistic insights into their modes-of-action remains a barrier to the development of new chemical probes and drug candidates. Xanthine-based modulators include the most potent and selective TRPC1/4/5 inhibitors described to date, as well as TRPC5 activators. Our previous studies suggest that xanthines interact with a, so far, elusive pocket of TRPC1/4/5 channels that is essential to channel gating. Here we report the structure of a small-molecule-bound TRPC1/4/5 channel—human TRPC5 in complex with the xanthine Pico145—to 3.0 Å. We found that Pico145 binds to a conserved lipid binding site of TRPC5, where it displaces a bound phospholipid. Our findings explain the mode-of-action of xanthine-based TRPC1/4/5 modulators, and suggest a structural basis for TRPC1/4/5 modulation by endogenous factors such as (phospho)lipids and $Zn^{2+}$ ions. These studies lay the foundations for the structure-based design of new generations of TRPC1/4/5 modulators.

[1] Discovery and Translational Science Department, Leeds Institute of Cardiovascular and Metabolic Medicine, University of Leeds, Leeds LS2 9JT, UK. [2] Astbury Centre for Structural Molecular Biology, University of Leeds, Woodhouse Lane, Leeds LS2 9JT, UK. [3] School of Biomedical Sciences, University of Leeds, Woodhouse Lane, Leeds LS2 9JT, UK. ✉email: s.p.muench@leeds.ac.uk; r.bon@leeds.ac.uk

Transient Receptor Potential Canonical (TRPC) proteins form homo- or heterotetrameric, non-selective cation channels permeable by $Na^+$ and $Ca^{2+}$[1–6]. Based on sequence similarity, the mammalian TRPC proteins can be further divided into sub-groups: TRPC1/4/5, TRPC3/6/7 and TRPC2[7], the latter of which is encoded by a pseudogene in humans[8]. TRPC4 and TRPC5 are the most closely related TRPC proteins[9] (70% sequence identity) and can form functional, homomeric TRPC4:C4 and TRPC5:C5 channels[10]. TRPC1 (48% and 47% sequence identity to TRPC4 and TRPC5, respectively) is not thought to form functional homomeric channels but is widely expressed and an important contributor to heteromeric TRPC1/4/5 ion channels[10–15].

Although disruption of the *Trpc4/5* genes[16] and global expression of a dominant-negative mutant TRPC5[12] do not cause catastrophic phenotypes in rodents, TRPC1/4/5 channels have been implicated in a wide range of physiological and pathological mechanisms[4,5,10,17]. These findings have driven the development of potent and selective TRPC1/4/5 modulators as chemical probes and drug candidates[5,10,18], and clinical trials have been started by Hydra Biosciences/Boehringer Ingelheim (the TRPC4/5 channel inhibitor BI 135889 for treatment of anxiety/CNS disorders) and Goldfinch Bio (the TRPC5 channel inhibitor GFB-887 for genetically driven kidney disease).

Physiological activation and modulation of TRPC1/4/5 channel activity is complex[4–6] and may include mediation by endogenous and dietary lipids[12,19–22] and metal ions such as $Zn^{2+}$[23]. In addition, structurally diverse pharmacological modulators have been reported[5,10,18], including inhibitors suitable for studies of TRPC1/4/5 in cells, tissues and animal models such as the xanthines Pico145 (also called HC-608)[24–26] and HC-070[25,26], the pyridazinone derivative GFB-8438[27], and the benzimidazole ML204[28]. However, structural insight into the mode-of-action of small-molecule TRPC1/4/5 modulators is lacking, and no small-molecule-binding sites have been identified.

Currently, the most potent and selective TRPC1/4/5 inhibitor is the xanthine Pico145, which inhibits the channels with $IC_{50}$ values in the picomolar to the low nanomolar range and displays the highest potency against heteromeric channels[24]. Pico145 is orally bioavailable and has been used successfully for studies of TRPC1/4/5 channels in vivo[26,29,30]. Detailed characterisation of Pico145 and the related xanthine AM237—a partial TRPC5 agonist that inhibits other TRPC1/4/5 channels[31]—revealed that: (1) Pico145 and AM237 act rapidly and reversibly in outside-out excised patch recordings, highlighting a membrane-delimited effect[24,31]; (2) Pico145 is a competitive antagonist of (−)-englerin A (EA)[24] and AM237[31], and also inhibits channel activation by sphingosine-1-phosphate (S1P), carbachol and $Gd^{3+}$[24,25], although picomolar concentrations of Pico145 can potentiate $Gd^{3+}$-induced TRPC4 currents as well[24]; (3) Pico145 concentration-dependently inhibits photoaffinity labelling of TRPC5 by the xanthine-based photoaffinity probes Pico145-DAAlk and Pico145-DAAlk2, with $IC_{50}$ values in the same range as inhibition of TRPC5-mediated calcium influx[32]. These results provide evidence that these xanthines modulate TRPC1/4/5 channels through direct molecular interaction with the channels and are consistent with the hypothesis that Pico145 and AM237 bind to a well-defined, high-affinity-binding site essential to TRPC1/4/5 channel gating. Targeting this binding site may be a promising strategy to develop further generations of potent and selective TRPC1/4/5 chemical probes and drug candidates.

Several high-resolution TRPC structures have been determined by single-particle cryo-electron microscopy (cryo-EM), including zebrafish TRPC4[33], mouse TRPC4[34] and mouse TRPC5[35]. To date, TRPC6 is the only TRPC channel for which small-molecule-bound structures have been published[36,37]. Here, we report the structure of a TRPC1/4/5 channel in complex with a small-molecule modulator: the homomeric TRPC5:C5 channel in complex with Pico145. We found that Pico145 binds between the *trans*-membrane domains of two TRPC5 subunits, where it displaces a phospholipid in a conserved lipid-binding site. Docking and mutagenesis studies support these findings and provide additional insights into the mode-of-action of xanthines as modulators of TRPC5 and TRPC4 channels. In addition, we have identified a putative intracellular zinc-binding site of TRPC5 that is conserved within the TRPC family. This work provides a rational basis for the design of new TRPC1/4/5 modulators and suggests important functional roles for the conserved TRPC1/4/5 lipid and zinc-binding sites. Therefore, our findings may help unravel physiological mechanisms of TRPC1/4/5 modulation, and pave the way for structure-based TRPC1/4/5 drug discovery efforts.

## Results

**Functional characterisation of C-terminally truncated human TRPC5.** For our structural studies, we engineered a human TRPC5 construct containing an N-terminal maltose-binding protein (MBP), followed by a PreScission protease site (PreS) and a C-terminally truncated (Δ766–975) hTRPC5 (99% identical to the analogous mTRPC5 construct[35]). Upon overexpression in HEK 293 cells, TRPC5:C5 channels formed by this construct were activated by EA ($EC_{50}$ 4.6 nM; cf. $EC_{50}$ 1.7 nM for full-length hTRPC5) and inhibited by Pico145 ($IC_{50}$ 4.0 nM; cf. $IC_{50}$ 1.8 nM for full-length hTRPC5) (Supplementary Fig. 1). These results suggest MBP-PreS-hTRPC5$_{Δ766–975}$ as a suitable construct for structural determination of the xanthine-binding site.

**Protein purification and quality control.** Baculoviruses were made for MBP-PreS-hTRPC5$_{Δ766–975}$ expression and used to transfect Freestyle™ 239-F Cells in suspension. MBP-tagged TRPC5 was purified using a protocol adapted from Duan et al.[35], but with amphipol (PMAL-C8) exchange performed on-resin and ultracentrifugation used in lieu of size exclusion chromatography to remove aggregated material. The protein was pure according to SDS-PAGE analysis (Supplementary Fig. 2a), and negative stain electron microscopy showed a monodisperse sample with particles consistent in size with the expected tetrameric structure (Supplementary Fig. 2b). Further negative stain analysis of the TRPC5 protein in the presence of Pico145 showed a similar monodispersity with no evidence for aggregation or disruption of the tetramer (Supplementary Fig. 2c).

**Structure of a human TRPC5:C5 channel in complex with Pico145.** The cryo-EM structure of a human TRPC5:C5 channel in the presence of Pico145 was determined with a global resolution of 3.0 Å using C4 symmetry (Fig. 1, Table 1 and Supplementary Fig. 3). To investigate whether there was heterogeneity between the four subunits, the complex was also refined with no symmetry imposed (C1 symmetry). The resulting TRPC5 structure showed no significant difference to the C4 refined model, but the global resolution was lower (3.3 Å). The tetrameric structure of TRPC5 was built (based on the mTRPC5 *apo* structure, PDB 6aei) and showed the typical TRPC fold: a large intracellular domain made up of the N- and C termini of each monomer, six *trans*-membrane helices per monomer and relatively short extracellular loops (Fig. 1b). The N terminus (residues 1–366) folds into 4 ankyrin repeats, a helical linker domain and the pre-S1-elbow domain. The first four *trans*-membrane helices fold into a voltage sensing-like domain (VSLD); the next two helices form the pore domain, together with the re-entrant pore helix (E3 loop), which lines the ion permeation pathway.

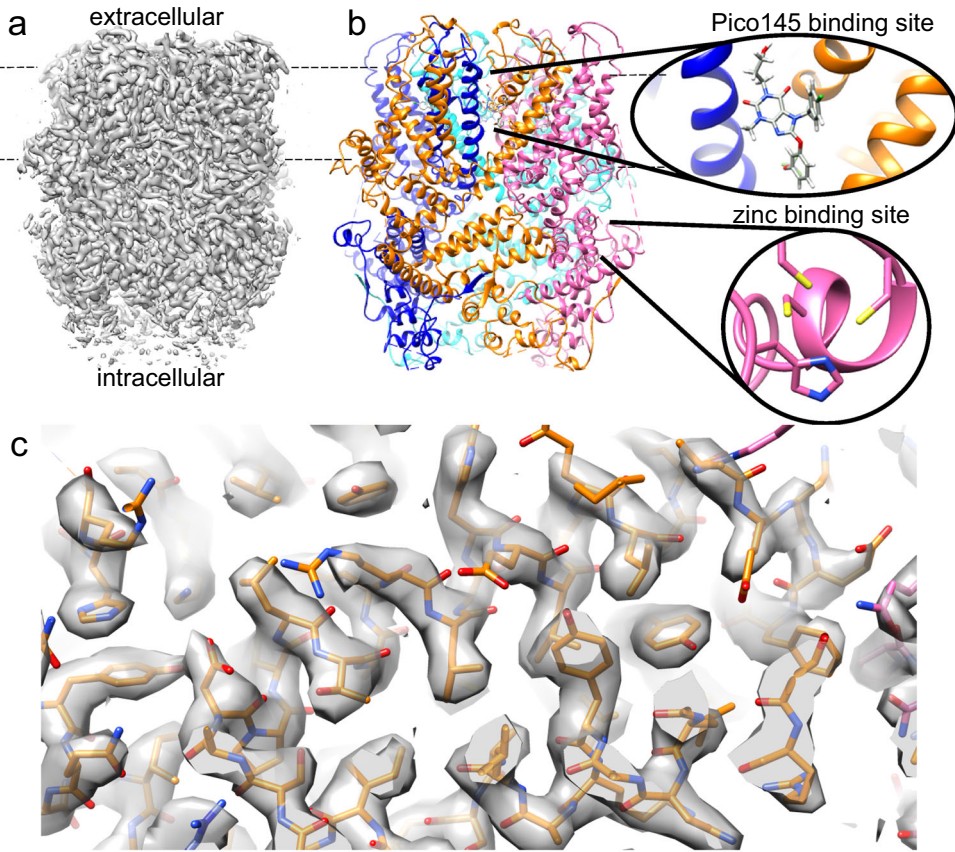

**Fig. 1 Structure of human TRPC5 in complex with Pico145. a** Cryo-EM density map of TRPC5:Pico145 (contour level: 0.0373). **b** Model of TRPC5:Pico145 coloured by monomer, showing the typical TRP channel domain swap architecture. Dashed lines indicate the predicted position of the lipid bilayer. Insets show one of the four Pico145-binding sites (between *trans*-membrane domains of the blue and orange TRPC5 monomers) and one of the four putative intracellular zinc-binding sites (in the magenta monomer). **c** Representative fit of TRPC5:Pico145 model **b** into the experimental density **a** showing helical regions spanning residues 193–234 (contour level: 0.0423).

The C terminus (residues 626 onwards) contains the TRP domain and the coiled-coil domain, which is formed by one helix from each monomer. The maltose-binding protein (MBP), protease site (PreS) and the first 16 amino acid residues of TRPC5 were not resolved in the map, likely because of flexibility. Likewise, residues 37–60, 74–95, 118–134, 273–285, 387–391, 665–702 and 734–765 were not modelled due to them being poorly resolved. All residues that were not modelled are found in the intracellular domain or extracellular loop 1; the *trans*-membrane helices and the other extracellular loops were modelled into the electron density.

Comparison of our TRPC5 structure to the previously reported 2.8 Å mTRPC5 *apo* structure (PDB 6aei)[35] revealed a backbone (Cα) RMSD of 0.81 Å showing a similar overall structure, consistent with both being in the closed state. There were, however, several notable local differences between the *apo* and Pico145-bound structures (Supplementary Fig. 4a, b). In our structure, the coiled-coil domain was less well ordered, and residues 734–759 could not be fit with confidence (and were therefore not included in our model). However, the quality of our map was such that additional residues could be built within the region 175–187, and that residues 172–174 could be placed with greater accuracy. This region showed that histidine 172 and cysteines 176, 178 and 181 all point towards a central density, consistent with metal ion binding (Figs. 1b and 5; see below for details). In addition, there was a notable difference in the non-protein density between the *apo* and Pico145-bound structures, as discussed below.

**Pico145 binds to a conserved lipid-binding site of TRPC5**. Comparison of the cryo-EM structure of TRPC5:Pico145 to published mTRPC5[35] and mTRPC4[34] *apo* structures showed clear differences in one of the lipid-binding sites (Fig. 2a–c). In the mTRPC5 *apo* structure, a density with a characteristic lipid U-shape was observed close to the E3 re-entrant loop (Fig. 2b). This density was ascribed to a phospholipid (postulated to be a phosphatidic acid or ceramide-1-phosphate)[35] that interacts with the LFW motif, which is conserved in the TRPC family (Supplementary Table 1). Indeed, a similar density is present in the mTRPC4 structure (Fig. 2c). The hTRPC3[36,38] and hTRPC6[36,37] structures contain lipids that interact with their LFW motifs as well, but with altered geometry (Supplementary Fig. 5). Our TRPC5:Pico145 structure also showed density in this region. However, the shape of this density was strikingly different to that in the mTRPC5 and mTRPC4 *apo* structures (Fig. 2a–c): in our structure, no density was observed for one of the lipid tails, and additional density was found between residues Q573, F569 and L572. Although this density was not consistent with a bound lipid, its shape and size were consistent with the presence of a molecule of Pico145. Indeed, modelling of Pico145 into this density gave an excellent fit (Fig. 3a). Similar to the phospholipid in the mTRPC5 and mTRPC4 *apo* structure, Pico145 (Fig. 3b) was observed between two TRPC5 monomers, with four molecules of Pico145 per TRPC5 tetramer. This was observed in both the C4 and C1 reconstructions and is consistent with all four sites being occupied.

**Table 1 Cryo-EM data collection, refinement and validation statistics.**

| | 2 mg ml$^{-1}$ hTRPC5 100 μM Pico145 (EMDB-10903) (PDB 6ysn) | 1 mg ml$^{-1}$ hTRPC5 50 μM Pico145 (EMDB-10909) | 1 mg ml$^{-1}$ hTRPC5 20 μM ZnCl$_2$ (EMDB-10910) |
|---|---|---|---|
| Data collection and processing | | | |
| Magnification | 130,000 | 130,000 | 130,000 |
| Voltage (kV) | 300 | 300 | 300 |
| Electron exposure (e-/Å$^2$) | 75 | 60 | 60 |
| Defocus range (μm) | −1 to −3 | −1 to −3 | −1 to −3 |
| Pixel size (Å) | 1.07 | 1.07 | 1.07 |
| Symmetry imposed | C4 | C4 | C4 |
| Initial particle images (no.) | 612,983 | 536,348 | 552,075 |
| Final particle images (no.) | 158,111 | 158,548 | 228,615 |
| Map resolution (Å) | 3.0 | 2.9 | 2.8 |
| FSC threshold | 0.143 | 0.143 | 0.143 |
| Map resolution range (Å) | 2.8–3.6 | 2.7–3.5 | 2.6–3.5 |
| Refinement | | | |
| Initial model used (PDB code) | 6aei | | |
| Model resolution (Å) | 3.0 | | |
| FSC threshold | 0.143 | | |
| Model resolution range (Å) | 2.8–3.6 | | |
| Map sharpening B factor (Å$^2$) | −100 | | |
| Model composition | | | |
| Non-hydrogen atoms | 19,700 | | |
| Protein residues | 2504 | | |
| Ligands | 4 | | |
| B factors (Å$^2$) | | | |
| Protein | 61.67 | | |
| Ligand | 22.44 | | |
| R.m.s. deviations | | | |
| Bond lengths (Å) | 0.005 | | |
| Bond angles (°) | 0.657 | | |
| Validation | | | |
| MolProbity score | 1.68 | | |
| Clashscore | 5.28 | | |
| Poor rotamers (%) | 0.19 | | |
| Ramachandran plot | | | |
| Favoured (%) | 94.07 | | |
| Allowed (%) | 5.93 | | |
| Disallowed (%) | 0.0 | | |

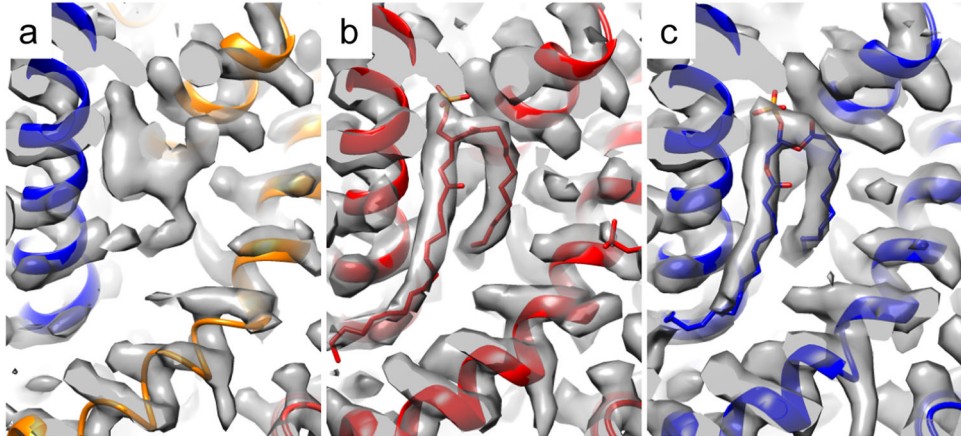

**Fig. 2 Pico145 binds to a conserved TRPC4/5 lipid-binding site. a** Density observed in our hTRPC5:Pico145 structure (100 μM Pico145; contour level: 0.0621). **b** Equivalent density in the published mTRPC5 *apo* structure (PDB 6aei, EMDB 9615) that was modelled as a phospholipid (contour level: 0.0510). **c** The same site in the published mTRPC4 structure (PDB 5z96, EMDB 6901; contour level: 0.0331).

Fitting Pico145 into the density shows that most of the TRPC5 residues interacting with the inhibitor are from helix S5 and the pore helix of the monomer on the right (monomer 1; Fig. 3a; orange). The 3-hydroxypropyl substituent on N-1 of Pico145 interacts with the side chains of W577 and Q573, apparently substituting for the phospholipid phosphate group in the *apo* structures. Pico145's xanthine core π-stacks with F576, and several residues (including L521, Y524, C525 and V579) interact with the 3-(trifluoromethoxy)phenoxy substituent on C8 of Pico145. The 4-chlorobenzyl substituent on N-7 of Pico145

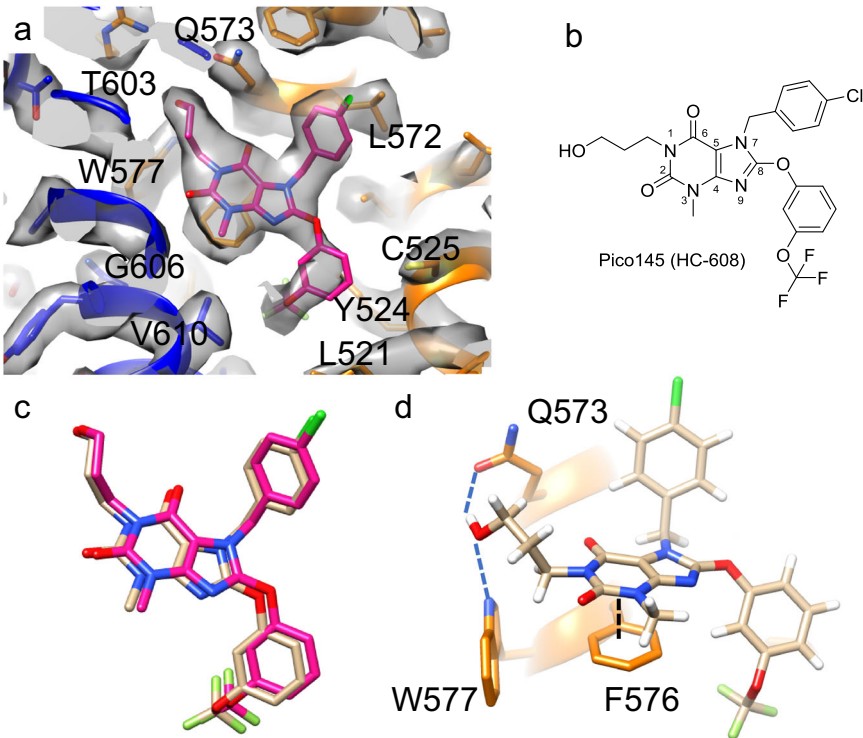

**Fig. 3 Analysis of the Pico145-binding site of TRPC5. a** The refined structure of the Pico145-binding site within the cryo-EM density map (grey; contour level 0.0583). The binding site spans two monomers: monomer 1 is shown in orange and monomer 2 is shown in blue; Pico145 is shown in magenta. Key residues in the binding site (L521, Y524, C525, L572, Q573, W577, T603, G606 and V610) are labelled. **b** Chemical structure of Pico145 with the numbering of the xanthine core. **c** An overlay of the top docking pose of Pico145 (peach) and our fitted Pico145 molecule (magenta). The top three docking poses are shown in Supplementary Fig. 7. **d** The top-scoring docking pose suggests that Pico145 hydrogen bonds to Q573 and W577 (dashed blue lines) and π-stacks with F576 (dashed black line).

points upwards, away from the lipid site, making a non-polar interaction with L572. Pico145 also makes additional interactions with helix S6 of the monomer on the left (monomer 2; Fig. 3a; blue): Pico145 interacts with T603 and packs against G606, and several hydrophobic amino acids (V610 and V614) interact with Pico145's 3-(trifluoromethoxy)phenoxy substituent. It should be noted that the local resolution of the cryo-EM map at the core of the TRPC5 channel, including the Pico145-binding site, was better than 2.9 Å, making the fit unambiguous (Supplementary Fig. 3).

During the optimisation of conditions for TRPC5:Pico145 cryo-grid preparation, we varied concentrations of TRPC5 and Pico145. As part of this process, an additional structure of TRPC5 in the presence of Pico145 was solved with TRPC5 at 1 mg ml$^{-1}$ and Pico145 at 50 μM (compared to 2 mg ml$^{-1}$ TRPC5 and 100 μM Pico145 in the structure described above) to a global resolution of 2.9 Å. The density around the Pico145-binding site in this structure showed features consistent with both the bound phospholipid and Pico145, suggesting partial occupancy under these conditions (Supplementary Fig. 6a–c). This observation may seem counter-intuitive, given the high potency of Pico145 and the fact that TRPC5:Pico145 ratios were identical in both samples. A reason could be that binding kinetics of Pico145 to purified TRPC5:C5 in amphipol may differ from those in the cellular context. These data further support the observation that Pico145 can displace each of the four phospholipids bound to a tetrameric TRPC5 channel.

**Docking of xanthines into TRPC5 and TRPC4 channel structures.** To further investigate the Pico145-binding site of TRPC5

we conducted docking experiments, using our TRPC5:Pico145 cryo-EM structure in which the coordinates for Pico145 were removed. Docking of Pico145 into a 36×36×36 Å grid around the Pico145-binding site produced three poses with similar predicted binding energies. These poses are almost identical to each other (Supplementary Fig. 7a) and to the pose of Pico145 modelled in our cryo-EM structure (Fig. 3c), giving confidence in the docking procedure and further validation to the observed density being assigned to Pico145. The lowest energy pose suggests that the 3-hydroxypropyl substituent at N-1 of Pico145 makes two hydrogen bonds, donating a hydrogen bond to the carbonyl of the side chain of Q573 and accepting a hydrogen bond from the indole of W577 (Fig. 3d and Supplementary Fig. 7b). This 3-hydroxypropyl substituent occupies a position similar to the phosphate head group of the phospholipid in other structures. Consistent with our TRPC5:Pico145 structure, all three poses show a π-stacking interaction between F576 and the xanthine core of Pico145 (Fig. 3d and Supplementary Fig. 7). Although the positively charged R557 is nearby, no significant interactions with this residue appear to take place. The xanthine core of Pico145 and the hydrophobic tails overlay almost perfectly in all three docked poses (Supplementary Fig. 7a). The main difference between poses is in the conformation of the 3-hydroxypropyl substituent, suggesting it is flexible in the TRPC5-binding pocket, where it may form hydrogen bonds with different TRPC5 residues. Structural alteration of the docked Pico145 in TRPC5 to the related xanthine AM237 (by addition of a chlorine atom) resulted in a potential clash with L521 (Supplementary Fig. 8), suggesting a structural rearrangement is required for AM237 to bind to TRPC5. This could explain why Pico145 is an inhibitor of TRPC5, whereas AM237 is a partial agonist [31].

Pico145 is also a potent inhibitor of other TRPC1/4/5 channels[24,25], including TRPC4:C4, for which a structure is available, and which has a lipid-binding site that is near-identical to that of TRPC5:C5 in terms of shape, bound phospholipid and residues lining the pocket (Supplementary Fig. 5b vs Supplementary Fig. 5c). Importantly, the TRPC5 residues predicted to interact with Pico145 are highly conserved in TRPC4, with the only difference being that V579 in TRPC5 is isoleucine in TRPC4 (Supplementary Table 1). In order to test if Pico145 would be able to bind in the lipid-binding site of TRPC4:C4, we next docked Pico145 into the published mTRPC4 *apo* structure (PDB 5z96; the phospholipid was removed before docking). The docking suggests that Pico145 adopts a similar pose in TRPC4 and TRPC5, making similar interactions with conserved residues, particularly residues equivalent to the TRPC5 residues Q573, F576 and W577 (Supplementary Fig. 9). These data suggest that the binding site of Pico145 is conserved between TRPC4 and TRPC5 channels.

**Mutations in the xanthine-binding site alter the response of TRPC5 to AM237.** TRPC4 and TRPC5 channels (including heteromeric TRPC1/4/5 channels containing TRPC1) are modulated by the xanthines Pico145 and AM237, whereas TRPC3/6/7 channels are not[24,25]. This is despite the high sequence similarity between TRPC proteins around the xanthine/lipid-binding sites of TRPC4 and TRPC5 channels (Supplementary Table 1). According to our structural modelling and docking, several residues were predicted to be important for binding of xanthines to TRPC5, and we decided to make and test three TRPC5 variants: Q573T, F576A and W577A (Fig. 3d and Table 2). Q573 is conserved between TRPC4 and TRPC5, but the analogous residues in TRPC1 and TRPC3/6/7 are phenylalanine and lysine, respectively (Supplementary Table 1). Q573T was chosen because threonine and glutamine are both hydrophilic amino acids with the propensity to form hydrogen bonds, but with altered geometries, which might affect xanthine binding. F576 and W577 are fully conserved TRPC residues that appear to interact with the co-purified lipid in TRPC3[36,38], TRPC6[36,37], TRPC4[34] and TRPC5[35] structures (Supplementary Fig. 5), and therefore could be important for function or molecular interactions. F576A should still be sufficiently hydrophobic to be stable in the membrane environment, but would not be able to π-stack with the xanthine core of Pico145/AM237. W577A would also retain the hydrophobic character at this position but would lose the ability of the W577 indole NH to act as a hydrogen bond donor to the 3-hydroxypropyl group of Pico145/AM237.

The constructs were tested using fluorometric intracellular calcium ($[Ca^{2+}]_i$) measurements (Fig. 4). Activation by EA[11] was used to show that TRPC5 variants were expressed and formed functional TRPC5:C5 channels. The inhibitory potency of Pico145 on TRPC1/4/5 channels is dependent on the EA concentration used in assays[24], which was expected to complicate the analysis of Pico145 responses of TRPC5 variants. Other commonly used TRPC5:C5 activators such as S1P and carbachol (at least partially) act through intracellular signalling, and may interfere with the composition of phospholipids bound to TRPC5,

while $Gd^{3+}$ has complex effects in the presence of Pico145[24]. Therefore, we decided to test TRPC5 responses to AM237, which is thought to occupy the same binding site as Pico145 (see above and refs. [31,32]). Mutations that resulted in TRPC5 activation by EA, but not by AM237, were likely to reveal residues contributing to xanthine binding. Wild-type TRPC5 was activated by EA ($EC_{50}$ 5.7 nM) and AM237 ($EC_{50}$ 10.6 nM) (Fig. 4a–c). Both EA ($EC_{50}$ 260 nM) and AM237 ($EC_{50}$ 110 nM) activated the TRPC5$_{Q573T}$ variant (Fig. 4d–f), but potencies of both activators were lower by at least an order of magnitude compared to wild-type TRPC5 (46-fold for EA and 10-fold for AM237; Table 2). TRPC5$_{F576A}$ (Fig. 4g–i) and TRPC5$_{W577A}$ (Fig. 4j–l) could be activated by EA, but again much higher EA concentrations were needed ($EC_{50}$ values of ≥1400 and 370 nM, respectively; Fig. 4i, l). However, neither TRPC5 variant could be activated by AM237 (no response up to 5 μM AM237; Fig. 4h, k), suggesting that F576 and W577 are key residues for the binding/gating of TRPC5 by AM237. Note that in wild-type HEK 293 cells, application of up to 5 μM of EA or AM237 did not result in increases in $[Ca^{2+}]_i$ (Supplementary Fig. 10), demonstrating that the signals detected in Fig. 4 are the result of TRPC5 channel activation rather than TRPC5-independent $Ca^{2+}$ handling processes. These results are consistent with our structural data and modelling/docking. In addition, the large decreases in EA potency seen in F576A (≥245-fold; Table 2) and W577A (65-fold; Table 2) may explain why Duan et al.[35] found that the mTRPC5 variant (F576A, W577A) was unresponsive to 100 nM EA.

**The TRPC5:Pico145 structure reveals a putative, conserved zinc-binding site.** Our TRPC5 cryo-EM map allowed residues 172–187 to be resolved and built. This region, which was not built into the previously published mTRPC5 *apo* structure[35], includes residues H172, C176, C178 and C181. The nature of the amino acids and the directionality of their thiol and imidazole substituents suggested the presence of a metal-binding site. A PDB motif query search, using hTRPC5 residues 172–181 (HXXXCXCXXC), revealed the presence of a similar binding site in the PHD domain of the zinc-binding E3 ubiquitin ligase UHRF1 (PDB 3zvz) where the motif coordinates a $Zn^{2+}$ ion. Manual overlay of this structure with our TRPC5 structure revealed a close alignment of residues (Fig. 5). The importance of this site is reflected by the complete conservation of these residues in the TRPC family (Supplementary Fig. 11). In addition, Park et al.[23] recently reported that intracellular application of $Zn^{2+}$ (micromolar concentrations) can open neuronal TRPC5 channels through an unknown mechanism. Therefore, we solved an additional structure of TRPC5 (1 mg ml$^{-1}$; ca. 7 μM of TRPC5 monomer) in the presence of $ZnCl_2$ (final concentration 20 μM) (Supplementary Fig. 3c). This structure, which was determined to a global resolution of 2.8 Å, showed that the addition of $ZnCl_2$ did not cause significant structural changes compared to our TRPC5:Pico145 structure described above. Again, the C-terminal coiled-coiled tail was more disordered than in the published mTRPC5 *apo* structure, suggesting that our slight alterations of the construct or purification protocol may be the cause of this disorder (Supplementary Fig. 4). The lipid/xanthine-binding site showed a

**Table 2 Summary of data in Fig. 4.**

| TRPC5 variant | Residue in TRPC4/1/3/6/7 | EA $EC_{50}$ (nM) | AM237 $EC_{50}$ (nM) |
|---|---|---|---|
| Wild-type | | 5.7 | 10.6 |
| Q573T | Q/F/K/K/K | 260 | 110 |
| F576A | F/F/F/F/F | ≥1400 | No response up to 5 μM |
| W577A | W/W/W/W/W | 370 | No response up to 5 μM |

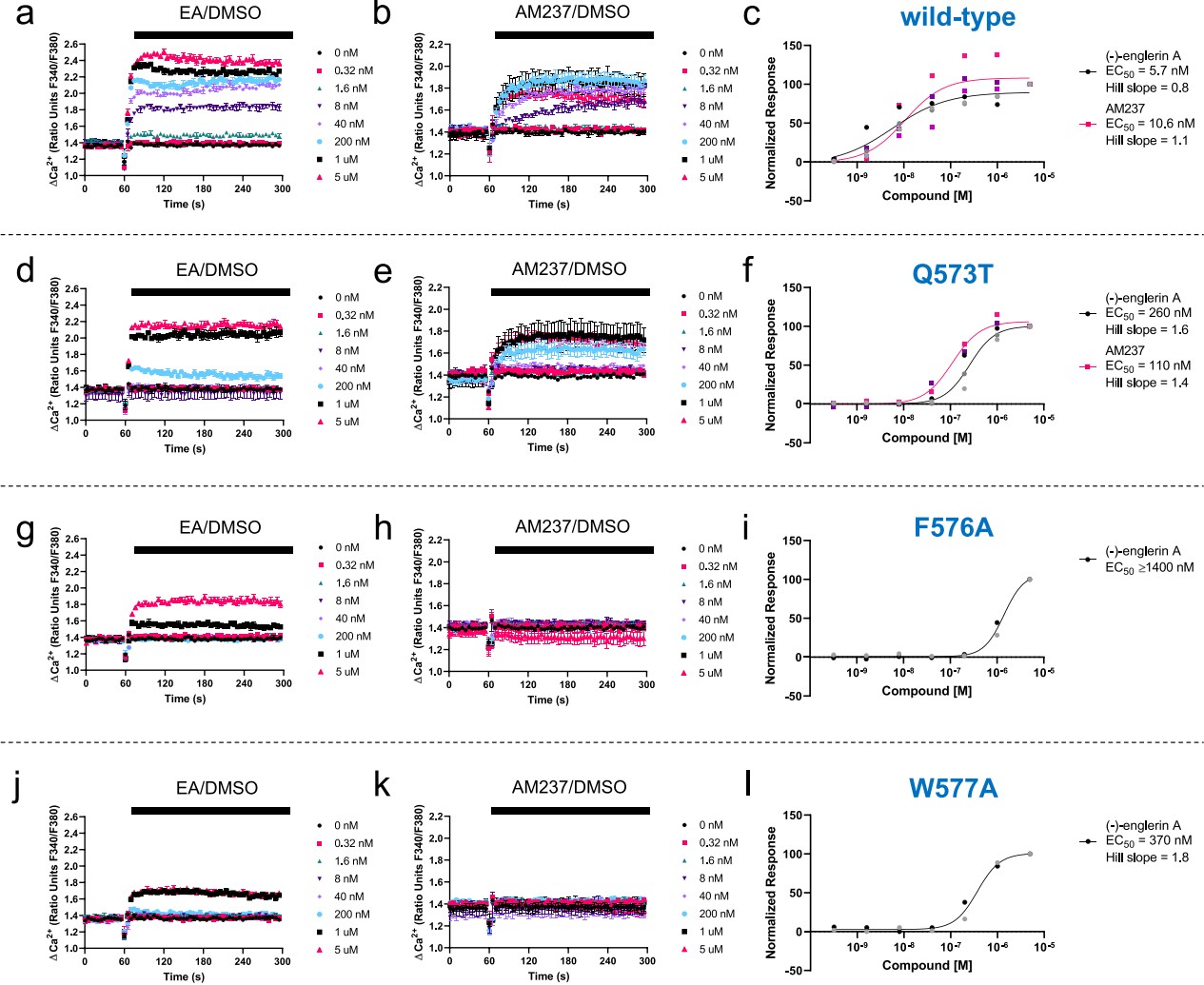

**Fig. 4 Mutations in the xanthine-binding site of TRPC5 affect responses to the xanthine-based TRPC5 agonist AM237. a**, **b**, **d**, **e**, **g**, **h**, **j**, **k** Representative traces from single 96-well plates (N = 3; error bars show standard deviations over technical replicates) showing changes of $[Ca^{2+}]_i$ in response to 0.32 nM to 5 μM of EA (**a**, **d**, **g**, **j**) or AM237 (**b**, **e**, **h**, **k**) in HEK 293 cells transiently expressing wild-type TRPC5-SYFP2 (**a**, **b**), TRPC5-SYFP2$_{Q573T}$ (**d**, **e**), TRPC5-SYFP2$_{F576A}$ (**g**, **h**) or TRPC5-SYFP2$_{W577A}$ (**j**, **k**). **c** Concentration–response data for experiments in **a** and **b** (scatter plots showing normalised data for three independent experiments each; n/N = 3/6). **f** Concentration–response data for experiments in **d** (scatter plot showing normalised data for three independent experiments; n/N = 3/6) and **e** (scatter plot showing normalised data for two independent experiments; n/N = 2/4). **i** Concentration–response data for experiments in **g** (scatter plot showing normalised data for two independent experiments; n/N = 2/4; because no upper limit of the EA response of F576A could be determined, the EC$_{50}$ for EA is unknown but can be estimated from the fitted curve to be at least 1400 nM; AM237 gave no response up to 5 μM). **l** Concentration–response data for experiments in **j** (scatter plot showing normalised data for two independent experiments; n/N = 2/5; AM237 gave no response up to 5 μM). Responses were calculated at 240–300 s compared to $[Ca^{2+}]_i$ at baseline (0–60 s). Averages of technical repeats of each individual experiment were normalised to the maximum response and vehicle control (0 nM) for the independent experiment, combined, and fit with GraphPad Prism 8 (variable slope, four-parameter logarithmic fit).

non-protein density consistent with a phospholipid (Supplementary Fig. 6a, c, d). In the zinc-binding domain, the densities were similar (and partially disordered) in the TRPC5:Zn$^{2+}$ and TRPC5:Pico145 structures. Because of this disorder, and the lower local resolution, we could not unambiguously identify the density of the Zn$^{2+}$ ion in either structure. These observations may suggest that this region is flexible and/or that fewer than four of the zinc-binding sites are occupied at any one time, even in the presence of added ZnCl$_2$. Interestingly, a structural alignment showed that the putative zinc-binding motif is present in the cryo-EM-derived structures of all TRPC channels; it was built into the hTRPC3[36,38] and hTRPC6[36,37] structures, but not into the mTRPC4[34] or mTRPC5[35] structures (Supplementary Fig. 11).

These findings suggest that TRPC channels have a conserved intracellular zinc-binding site.

## Discussion
The widely expressed TRPC1/4/5 channels have been implicated in various physiological and pathological mechanisms and are an emerging class of potential drug targets[4,5,10,17]. Our study provides structural insight into small-molecule modulation of a TRPC1/4/5 channel. By determination of a 3.0 Å structure of the TRPC5:C5 channel in complex with the TRPC1/4/5 inhibitor Pico145, we show that Pico145 can bind to up to four lipid-binding sites between TRPC5 subunits, where it displaces

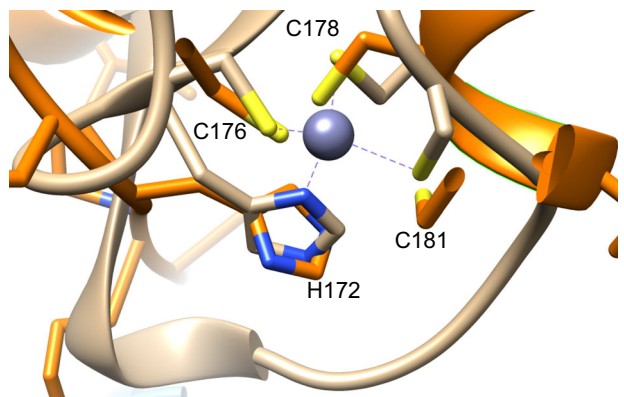

**Fig. 5 Structure alignment of hTRPC5 and hURF1 is consistent with the identification of a TRPC5 zinc-binding site.** Sequence alignment revealed that the $Cys_3$-$His_1$ motif closely matched that of the PHD finger of human UHRF1 (an unrelated zinc-binding protein). Manual alignment was performed between hTRPC5 (orange) and hUHRF1 (PDB 3zvz; wheat). hTRPC5 residues forming the putative zinc-binding site have been annotated.

phospholipids that interact with the channel's pore helices, previously proposed to be ceramide-1-phosphate or a phosphatidic acid[35]. Pico145 could be fit unambiguously into the non-protein density between TRPC5 subunits, which was strikingly different from the density in the corresponding sites of the published mTRPC5[35] and mTRPC4[34] *apo* structures, and of our 2.8 Å structure of TRPC5 determined in the presence of $ZnCl_2$. Our findings were further corroborated by a partially-occupied TRPC5:Pico145 structure (showing density for both Pico145 and the phospholipid), molecular docking, and site-directed mutagenesis. We hypothesise that Pico145 initially integrates with the lipid membrane, before entering the lipid-binding site between two TRPC5 subunits and replacing the bound phospholipid. This hypothesis is consistent with the hydrophobic nature of Pico145, and with the observation that the effect of Pico145 is not as rapid as would be expected for a pore blocker [24].

The xanthines Pico145 and HC-070 are the most potent and selective TRPC1/4/5 inhibitors to date[24,25]. However, three closely related xanthine derivatives, AM237[31] and the photoaffinity probes Pico145-DA and Pico145-DAAlk[32], are partial agonists of the homomeric TRPC5:C5 channel that inhibit other TRPC1/4/5 channels. Pico145 is a competitive antagonist of AM237[31] and concentration-dependently inhibits both activation and photo-affinity labelling of TRPC5:C5 by Pico145-DAAlk[32]. These data are consistent with the hypothesis that xanthine-based TRPC1/4/5 modulators all bind to the same lipid-binding site of TRPC1/4/5 channels (this paper and refs. [31,32]), where they can stabilise either open or closed channel conformations depending on xanthine substituent patterns and the composition of the specific TRPC tetramer. Therefore, targeting the xanthine-binding site of TRPC1/4/5 channels may allow the development of modulators of specific TRPC channel tetramers, a major challenge in TRPC1/4/5 pharmacology.

TRPC1/4/5 channels are modulated by endogenous and dietary lipids, but the mechanisms are complex and incompletely understood[12,19–22]. Lipids may affect TRPC1/4/5 channels indirectly, e.g., the phospholipid S1P can activate channels (at least in part) through G-protein signalling[39]. However, more direct mechanisms are possible as well, e.g., lysophosphatidylcholine (LPC) activates TRPC5 in excised outside-out patches in the absence of GTP[40]. Such mechanisms may involve direct molecular interactions and/or changes to the membrane environment of TRPC1/4/5 channels. The TRPC5 lipid-binding

site that Pico145 interacts with is highly conserved within the TRPC family (Supplementary Table 1). Bound (phospho)lipids have been found in this site in structures of hTRPC3[36,38], mTRPC4[34], mTRPC5[35] and hTRPC6[36,37], and the TRPC6 activator AM-0883 displaces the ordered lipid bound in this site near the P-loop of hTRPC6 (Supplementary Fig. 5f)[37]. The hTRPC5 residues F576 and W577 are conserved throughout the TRPC family and are likely to be responsible for lipid binding. Mutations of these residues resulted in TRPC5 variants that displayed lower sensitivity to EA and lacked a response to AM237. This suggests that EA may bind in the lipid/xanthine-binding site as well, which is consistent with the observation that the potency of Pico145 is dependent on the concentration of EA used in assays[24]. Our study suggests that binding of (phospho)lipids to the conserved TRPC lipid-binding sites is a dynamic, reversible process that may have a direct functional effect on channel gating, and that this site can be targeted by TRPC1/4/5 activators as well as inhibitors. However, further studies are required to fully understand the mechanisms of TRPC1/4/5 channel binding and gating by (phospho)lipids and different small-molecule TRPC1/4/5 activators such as EA.

Intracellular $Zn^{2+}$-dependent activation of TRPC5 channels was recently reported to contribute to oxidative neuronal death[23], but the molecular mechanism of $Zn^{2+}$ regulation of TRPC5 was not determined. We have identified a putative intracellular zinc-binding site of TRPC5, which is conserved in all TRPC channels. Cryo-EM of TRPC5 in the presence of $ZnCl_2$ resulted in a further TRPC structure determined to a resolution of 2.8 Å. Presence of $ZnCl_2$ did not change the overall structure compared to TRPC5: Pico145, and the density of the $Zn^{2+}$ ion could not be unambiguously identified in either structure. These observations suggest flexibility of the region or partial maximum occupancy of the zinc-binding site, and may indicate that zinc is already bound in TRPC5:Pico145, or that the role of zinc in TRPC5 modulation is subtle. A subtle role of $Zn^{2+}$ in TRPC5 modulation would also be consistent with the delayed $Zn^{2+}$-mediated $[Ca^{2+}]_i$ increase observed in calcium imaging experiments, and the small TRPC5 currents evoked by intracellular application of $ZnCl_2$[23]. Part of the putative zinc-binding site has also been implicated in TRPC5 glutathionylation: mutation of C176, C178 or C181 prevented TRPC5 opening in response to glutathionylation[41]. A separate report proposed that palmitoylation of C181 is required for correct trafficking of TRPC5 to the plasma membrane in striatal neurons[42]. These data suggest that this small domain may be an important regulatory node of TRPC channels and that further experiments are needed to fully understand its role in TRPC channel biology.

To the best of our knowledge, our data provide the first structural insight into TRPC1/4/5 channel modulation and suggest direct modulatory roles for (phospho)lipids and $Zn^{2+}$ ions. Our data are consistent with the description of a structure of hTRPC5 in complex with the xanthine HC-070 in a non-peer-reviewed study by Song et al.[43], which appeared shortly after the initial disclosure of our study on bioRxiv. These studies lay the foundations for the structure-based design of TRPC1/4/5 modulators, and may therefore support the development of new TRPC1/4/5 chemical probes and drug candidates for an increasing number of therapeutic areas.

## Methods

**Plasmids**. All plasmids were cloned using NEBuilder HiFi DNA Assembly Master Mix (New England Biolabs) and mutations were made using the Q5 Site-directed mutagenesis kit (New England Biolabs). The BacMam vector was a kind gift from Professor Eric Gouaux (Vollum Institute)[44]. Briefly, human TRPC5 was cloned as a C-terminally truncated form (Δ766–975) with an N-terminal maltose-binding protein tag followed by a PreScission protease cleavage site (as in ref. [35]) using TRPC5-SYFP2[31] as a PCR template for TRPC5, which contains the previously

described V455M mutation[45]. Point mutations to the xanthine-binding site were introduced into TRPC5-SYFP2 in pcDNA4. Bacmids and baculoviruses were produced according to the Bac-to-Bac protocol (Invitrogen).

**Protein expression and purification.** P2 virus was added to 2.0 million per ml of Freestyle™ 293-F Cells (ThemoFisher Scientific) in Gibco FreeStyle 293 Expression Medium (Invitrogen) at a final volume of 10% at 37 °C and 5% $CO_2$. After 24 h, 5 mM sodium butyrate (Sigma Aldrich) was added and the temperature was lowered to 30 °C. After a further 48 h, cells were harvested by centrifugation and frozen. The protein purification protocol was adapted from Duan et al.[35]. Unless stated otherwise, all detergents (and amphipol PMAL-C8) were supplied by Generon. For a typical purification, a 200 ml cell pellet was thawed and resuspended in 20 ml of 1% DDM, 0.1% CHS, 150 mM NaCl (Sigma Aldrich), 30 mM HEPES (Sigma Aldrich) pH 7.5, 1 mM DTT (Fisher Scientific Ltd) and protease inhibitor cocktail (Sigma Aldrich), and incubated by rotating at 4 °C for 1 h. After ultracentrifugation at 100,000×$g$ for 1 h at 4 °C, the supernatant was incubated with 400 µl bed volume of pre-washed amylose resin (New England Biolabs) for 18 h, rotating at 4 °C. The resin was washed with 20 ml of 0.1 % digitonin and 0.01% CHS, 150 mM NaCl, 30 mM HEPES pH 7.5, 1 mM DTT. The resin was resuspended in 10 ml of 0.1% PMAL-C8, 150 mM NaCl, 30 mM HEPES pH 7.5, 1 mM DTT and rotated for 4 h at 4 °C. 100 mg of Biobeads (Bio-rad) were then added and incubated for 18 h, rotating at 4 °C, to remove detergents. The resin was washed with 20 ml of buffer containing no detergent (150 mM NaCl, 30 mM HEPES pH 7.5, 1 mM DTT) before eluting in 5 ml of the same buffer plus 40 mM maltose (Sigma Aldrich). The eluate was subjected to ultracentrifugation at 100,000×$g$ for 1 h at 4 °C to remove any precipitated material and the supernatant was concentrated to the required concentration with 100 kDa cut off vivaspin 500 concentrators (Sigma Aldrich). In a typical purification protocol, 50–100 µg of purified TRPC5 was produced from 200 ml of cell suspension.

**Negative stain electron microscopy.** Sample quality was assessed by negative stain electron microscopy. Briefly, carbon-coated grids were glow-discharged for 40 s in a Pelco glow discharge unit. After glow-discharging, 3 µl of purified TRPC5 at 0.04 mg ml$^{-1}$ was added to the grid and the sample was stained as follows[46]. The grid was incubated with TRPC5 at room temperature for 30 s and the excess protein solution was wicked away using filter paper (Whatman Grade 1). This was followed by two successive incubations with uranyl acetate stain, in each of which 3 µl of 1% uranyl acetate was applied to the grid, followed by 1 min incubation and wicking away of excess liquid. The grid was then left to air dry. For Pico145-containing grids, Pico145 (10 mM in DMSO) was added to 2 mg ml$^{-1}$ purified TRPC5 to a final concentration of 100 µM, which was subsequently diluted 50-fold in detergent-free wash buffer, resulting in a final concentration of 2 µM Pico145 and 0.04 mg ml$^{-1}$ TRPC5. Negative stain grids were imaged using a Technai T12 microscope fitted with a LaB6 filament operating at 120 kV with a nominal magnification of ×49,000 on a 2 k × 2 K Gatan CCD camera.

**Sample preparation and data collection.** Pico145 (final concentration of 100 µM, from a 10 mM DMSO stock; final DMSO concentration 1%) was added to purified TRPC5 in PMAL-C8 at 2.0 mg ml$^{-1}$. A 3 µl aliquot of the sample was applied to a Quantifoil Cu R1.2/1.3, 300 mesh holey carbon grid, which had been glow-discharged for 30 s using a Pelco glow discharge unit. An FEI Vitrobot was used to blot the grids for 6 s (blot force 6) at 100% humidity and 4 °C before plunging into liquid ethane. The grids were loaded into an FEI Titan Krios transmission electron microscope (Astbury Biostructure Laboratory, University of Leeds) operating at 300 kV, fitted with a Gatan K2 direct electron detector operating in counting mode. Automated data collection was carried out using EPU software (2.2) and Digital Micrograph (3.23) with a defocus range of −1 to −3 µm. A total of 3,482 micrographs were collected with a pixel size of 1.07 Å. The total dose, 75 e− Å$^{-2}$, was acquired by use of a dose rate of 8.64 e− pixel$^{-1}$ s$^{-1}$ (7.58 e− Å$^{-2}$ s$^{-1}$) across 40 frames in 10 s total exposure time. The parameters for the structures in the presence of $ZnCl_2$ or 50 uM Pico145 used 1 mg ml$^{-1}$ final protein concentration and used the same EPU software with a defocus range of −1 to −3 µm. Micrographs were collected with a pixel size of 1.07 Å. The total dose, 60 e− Å$^{-2}$, was acquired by using a dose rate of 8.54 e− pixel$^{-1}$·s$^{-1}$ (7.49 e−Å$^{-2}$ s$^{-1}$) across 40 frames for 8 s total exposure time.

**Image processing.** An overview of the image processing protocol is shown in Supplementary Fig. 12. All processing was completed in Relion-3.0.7 unless stated otherwise[47]. The initial drift and beam-induced motion were corrected for using MotionCor2[48], and CTf estimation was performed using Gctf[49]. Automated particle picking was performed in crYOLO (1.3.5)[50] using the general model. For the Pico145-bound structure, this resulted in a particle stack containing 612,983 particles that were imported into Relion-3.0.7. An initial round of 2D classification resulted in a particle stack consisting of 519,962 particles. A 3D initial model was generated in Relion and low-pass filtered to 60 Å when used in 3D classification. The best class, consisting of 266,887 particles, was taken forward and refined with C4 symmetry imposed to give a global resolution of 3.3 Å after post-processing, with resolutions estimated by the gold standard FSC = 0.143 criterion. Three iterative rounds of CTF refinement[51] and particle polishing were completed, which

improved the resolution of the map to 3.1 Å. The polished particles underwent two further rounds of 3D classification resulting in a particle stack containing 158,111 particles that were refined to 3.0 Å. A local resolution map was calculated in Relion which showed that the core of the structure was at a higher resolution (2.8 Å) than the global average. For the structure in the presence of $ZnCl_2$, crYOLO picked 552,075 particles, which was reduced to 479,810 upon 2D classification. 3D classification further reduced this to 228,615 particles, which were refined and post-processed, resulting in a 3.1 Å structure. After three iterative rounds of CTF refinement and particle polishing, the resolution of the map was improved to 2.8 Å —further 3D classification did not improve the map. For the 50 µM (partial occupancy) Pico145 structure, micrographs were subjected to crYOLO particle picking (536,348 particles), followed by 2D classification (509,928 particles), 3D classification (158,548 particles). 3D classified particles were subjected to refinement and post-processing, resulting in a 3.1 Å structure. Three rounds of CTF refinement and particle polishing improved the overall resolution to 2.9 Å—further 3D classification did not improve the map.

**Creation of PDB file.** The model of TRPC5 with 100 µM Pico145 was produced by the manual fitting of 6aei into the model. Several rounds of real-space refinement in Phenix (1.16 −3549)[52] were performed before fitting Pico145 into the map in WinCoot (0.8.9.2)[53]. Models and maps were visualised and figures prepared in UCSF Chimera (1.13.1)[54].

**Docking studies.** The region of the TRPC5 tetrameric structure assigned for docking studies was chosen based upon residues identified to be close Pico145 in our TRPC5:Pico145 structure (Y524, L572, Q573, F576, W577, V579, F599, Y603 and V614). A 25 Å clip of the TRPC5 structure around these residues was termed as the receptor for docking studies using Glide (Schrödinger Release 2019-4, Glide, Schrödinger, LLC, New York, NY, 2020)[55]. The TRPC5 pdb file was prepared using the Protein Preparation Wizard in the Maestro Graphical User Interface (GUI). This aimed to remove any steric clashes of amino acid side chains and optimise the position of hydrogen atoms to facilitate docking studies. The receptor grid was generated using Schrödinger software, allowing docking of ligands in a 36×36×36 Å grid. The Pico145 ligand was prepared using the OMEGA module[56] of OpenEye software (OMEGA version 2.5.1.4 OpenEye Scientific Software, Santa Fe, NM; http://www.eyesopen.com) to produce an energy-minimised 3D structure before importing into the Maestro GUI. Docking of Pico145 was carried out using the Glide module of Schrödinger software using the XP mode with flexible ligand sampling and biased sampling of torsions for all predefined functional groups. Epik state penalties were added to the docking score. A maximum of 10 poses for the ligand was requested in the output file and post-docking minimisation was carried out.

**Intracellular $Ca^{2+}$ measurements.** Intracellular calcium ([$Ca^{2+}$]$_i$) was measured using the ratiometric $Ca^{2+}$ dye Fura-2. HEK 293 cells (ATCC, Teddington, UK) were plated in 6-well plates at 1 million cells per well. The following day, cells were transfected with 2 µg DNA using 6 µl of JetPrime (VWR International) according to the manufacturer's protocol. The next day cells were washed, trypsinised and plated onto black, clear-bottom poly-D-lysine coated 96-well plates with 60,000 cells per well. 24 h after plating, the cells were loaded with the Fura-2 dye by removal of media and addition of SBS (NaCl 130 mM, KCl 5 mM, glucose 8 mM, HEPES 10 mM, $MgCl_2$ 1.2 mM and $CaCl_2$ 1.5 mM; all supplied by Sigma Aldrich) containing 2 µM Fura-2 acetoxymethyl ester (Fura-2 AM; ThermoFisher Scientific) and 0.01% pluronic acid (Merck) for 1 h at 37 °C. After this incubation, cells were washed with fresh SBS and incubated at room temperature for a further 30 min. SBS was then changed to recording buffer (SBS with 0.01% pluronic acid and 0.1% DMSO to match compound buffer) immediately prior to experimentation. In the case of inhibitor studies, the buffer was replaced with SBS with 0.01% pluronic acid and the relevant inhibitor or vehicle. Cells were then incubated for 30 min prior to the experiment. Measurements were carried out using a FlexStation and SoftMax Pro 7 software (Molecular Devices, San Jose, CA), using excitation wavelengths of 340 and 380 nm, at an emission wavelength of 510 nm. ([$Ca^{2+}$]$_i$) recordings was performed at room temperature at 5 s intervals for 300 s (unless stated otherwise). EA or AM237 was added by the FlexStation from a compound plate (final DMSO concentration 0.1%) after recording for 60 s. Responses were calculated at 240–300 s compared to the signal at baseline (0–60 s), and an average was taken within each individual experiment. Averages from individual experiments were normalised relative to the maximum signal for each TRPC5 variant/activator combination and to the vehicle control, combined and fit (using a variable slope, 4-parameter logarithmic fit) with GraphPad Prism 8 with [agonist] for EA or AM237, or [inhibitor] for Pico145 vs response.

**Statistics and reproducibility.** For intracellular calcium recordings, $n$ (the number of independent experiments, i.e. different plates with different batches of cells) and $N$ (the total number of replicates from $n$ independent experiments) have been defined in the figure legends.

**Chemicals.** Pico145[24] and AM237[31] were prepared according to previously reported procedures. (−)-englerin A (EA) was obtained from PhytoLab

(Vestenbergsgreuth, Germany). AM237, Pico145 and EA were made up as 10 mM stocks in 100% DMSO, aliquots of which were stored at −20 °C (AM237 and Pico145) or −80 °C (EA). Further dilutions of compounds were made in DMSO and these were dissolved 1:1000 in a compound buffer (SBS + 0.01% pluronic acid) before being added to cells. Fura-2 AM (Invitrogen UK) was dissolved at 1 mM in DMSO.

**Reporting summary**. Further information on research design is available in the Nature Research Reporting Summary linked to this article.

## Data availability

Protein sequences and structural data are available via the Protein Data Bank (PDB 6YSN) and the Electron Microscopy Data Bank (EMDB-10903, EMDB-10909 and EMDB-10910). Raw data for intracellular calcium recordings (Fig. 4) are available in Supplementary Data 1.

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

## Acknowledgements
This work was supported by the BBSRC (BB/P020208/1) and a Wellcome Trust PhD studentship (102174/B/13/Z) to R.M.J. Professor Eric Gouaux (Vollum Institute) is kindly acknowledged for providing the BacMam vector. Large-scale tissue culture was performed in the University of Leeds Protein Production Facility (funded by the University of Leeds and the Royal Society), with support provided by Dr Brian Jackson. We thank the Astbury Biostructure Laboratory (funded by the University of Leeds and the Wellcome Trust) for support of electron microscopy work, Negative stain microscopy was performed using a T12 microscope funded by the Wellcome Trust (090932/Z/09/Z). Cryo-EM was performed with a Titan Krios microscope with an energy-filtered Gatan K2 XP summit direct electron detector (108466/Z/15/Z).

## Author contributions
D.J.W. performed the design and cloning of constructs, intracellular calcium measurements and protein purification. D.J.W., R.M.J. and S.P.M. performed electron microscopy studies, including data analysis and model building. K.J.S. performed docking studies. D.J.W., K.J.S., R.M.J., S.P.M. and R.S.B. analysed data. D.J.B., S.P.M. and R.S.B. conceived the project and generated research funds. S.P.M. and R.S.B. led the project. D.J.W., K.J.S., S.P.M. and R.S.B. prepared figures and wrote the manuscript. All authors commented on the manuscript.

## Competing interests
The authors declare the following competing interests: David J. Beech is an inventor on the following patent applications: (1) PCT/GB2018/050369. TRPC ion channel inhibitors for use in therapy. Filing date: 9th February 2018. Inventors: David J. Beech, Richard J. Foster, Sin Ying Cheung and Baptiste M Rode; (2) 62/529,063. Englerin derivatives for the treatment of cancer. Filing date: 6th July 2017. Inventors: John A. Beutler, Antonio Echavarren, William Chain, David Beech, Zhenhua Wu, Jean-Simon Suppo, Fernando Bravo and Hussein Rubaiy.
