## [Peer Review File · Communications Biology]

Reviewers' comments:

Reviewer #1 (Remarks to the Author):

"Cryo-EM structures of human TRPC5 reveal interaction of a xanthine-based TRPC1/4/5 inhibitor with a conserved lipid binding site" by David J. Wright et al, reported the first 3.0 Å cryoEM structure of truncated human TRPC5 homotetramer complexed with Pico-145, a xanthine modulator that is an inhibitor to TRPC5 ion channel. The overall structure of TRPC5-Pico-145 complex is similar to that of published mTRPC5 (Duan, J. et al. *Sci. Adv.* 5, eaaw7935, 2019), except the phospholipid was replaced by Pico-145. Fitting the Pico-145 structure into the corresponding no-protein density give an excellent fit. The authors also run the docking procedure and performed the site-directed mutagenesis experiment to confirm the binding site with specific amino acids. The data is solid and the cryoEM structure of hTRPC4-Pico145 complex is decent. The findings from this work provide the foundations for the structure-based design of new generations of TRPC1/4/5 modulators. However, it would be better and more convincing for authors to use Pico145, but not the AM237 in testing the response of TRPC5 with mutations in putative binding site for two reasons: first the Pico145 and Am235 is not exactly same in terms of modulating the function of TRPC5, and secondly the Pico145 was used in this study to form the TRPC5-Pico145 complex.

Following are some specific questions/comments for authors:

1. Page3, line3-5 "Further negative stain analysis of the TRPC5 protein in the presence of Pico145 showed a similar monodispersity with no evidence for any gross conformational changes or disruption of the tetramer." The negative images of TRPC5 tetramer with/without Pico-145 (Supplementary Figure 2, panel B, C) do looks different interms of staining and appearance of particles. Please explain the difference.
2. Figure 1: please label the Pico-145 and putative zinc binding sites with clear text, something like "Pico145 binding site" or "Zinc binding site".
3. Does the reference map used in the 3D reconstruction of TRPC5 without symmetry imposed come from the 3D reconstruction that has symmetry imposed?
4. Page4: "Comparison of our TRPC5 structure to the previously reported 2.9 Å mTRPC5 apo structure" should be "Comparison of our TRPC5 structure to the previously reported 2.8 Å mTRPC5 apo structure"
5. In figures 2, 3(panel A), Supplementary figures 5, 6, and 10, the atomic models have the same colors as their corresponding EM density map, which gave the poor contrast and make them hard to see, this is especially bad for grey atomic models in grey density maps. Suggest to adjust the color to make the figures with better contrast.
6. Page 5, First paragraph about the Figure 3a. In addition to W577, W573 and F576, there are many other amino acid side chains also involved in the interaction with Pico145, as described in the text (L521, Y524, C525, V579, L572, T603, G606, V610, V614), please also label these amino acids in the figure 3a to make them clear for general readers. This should also apply to Supplementary figure 7b (page 22).
7. Page 5, last paragraph: "During the optimisation of conditions for TRPC5:Pico145 cryo-grid preparation, we varied concentrations of TRPC5 and Pico145. As part of this process, an additional structure of TRPC5 in the presence of Pico145 was solved with TRPC5 at 1 mg·ml⁻¹ and Pico145 at 50 μM (compared to 2 mg·ml⁻¹ TRPC5 and 100 μM Pico145 in the structure described above) to a global resolution of 2.9 Å. The density around the Pico145 binding site in this structure showed features consistent with both the bound phospholipid and Pico145, suggesting partial occupancy under these conditions (Supplementary Figure 6A-C). These data further support the observation that Pico145 can displace each of the four phospholipids bound to a tetrameric TRPC5 channel." The sample 2 mg·ml⁻¹ TRPC5 with 100 μM Pico145 and the sample of 1mg·ml⁻¹ TRPC5 with 50 μM Pico145 should have same molar ratio of TRPC5-Pico145, but the 2.9 Å cryoEM map from later suggested the partial occupancy of Pico145. Please explain.
8. Page 8, figure 4 and Page 23, supplementary figure 8, please label the amino acids that are

involved in the interactions.

9. Page 6 : "To gain insight into the potent inhibition of TRPC4 channels by Pico14523, we next docked Pico145 into the published mTRPC4 apo structure (PDB 5z96; the phospholipid was removed before docking). The docking suggests that Pico145 adopts a similar pose in TRPC4 and TRPC5, making similar interactions with conserved residues, particularly residues equivalent to the TRPC5 residues Q573, F576 and W577 (Supplementary Figure 9). These data suggest that the binding site of Pico145 is conserved between TRPC4 and TRPC5 channels."

Because the published mTRPC5 and mTRPC4 structure are almost identical, especially in the region where phospholipid binds, but their physiological functions do differ. Therefore, by just docking the Pico145 into the mTRPC4 structure without other supportive data, it's pretty risky to draw such a conclusion.

10. Page 8: authors claim their TRPC5:Pico145 structure reveals a putative, conserved zinc binding site. If there is any EM density corresponding to this bound Zinc? Authors used sequence alignment of hTRPC5 and hURF1 to support their point, but at resolution of 2.8 Å with the presence of ZnCl₂, there should be an EM density showing the Zinc is indeed exist.

11. Page 10, line 5: "the highest resolution TRPC5 structure to date." This is not true, because ref. 34 "Cryo-EM structure of TRPC5 at 2.8 Å resolution reveals unique and conserved structural elements essential for channel function" by Duan et al reported a 2.8 Å TRPC5 structure already.

Reviewer #2 (Remarks to the Author):

The manuscript by Wright and colleagues reports cryo-EM structure of human TRPC5 in complex with small-molecule Pico145, which modulates channel function by occupying a common phospholipid binding site conserved in TRPC channels. The assignment of the inhibitor in the structure is supported by molecular docking and functional analysis with site-directed mutagenesis. In addition, a zinc-binding site has been identified. The cryo-EM structures seem to be of good quality and interpretation of structural data is reasonable. The manuscript improves our understanding of TRPC structure and function. The following points, mainly structural illustrations, need to be addressed before the manuscript is ready for publication.

Structural illustrations throughout the manuscript need to be improved for clarity. For instance, all structural illustrations with cryo-EM density maps were not of high quality. The contour levels were not specified. Structural elements and important residues mentioned in the text were not labeled in the figures. In addition, segmented cryo-EM density maps should be shown for assessment of local map quality.

The authors stated that "We tentatively modelled residues 734 to 759, yet the position of these residues should be interpreted with caution". Do the authors mean that this region was not confidently built? Maybe consider poly-Ala model for this region. Again, EM density should be shown.

The authors stated that "This region showed that histidine 172 and cysteines 176, 178 and 181 all point towards a central density, consistent with metal ion binding (Figure 1B and Figure 5; see below for details)". Cryo-EM density for Zn²⁺ was not shown.

Fig. 3 is supposed to show detailed interactions of Pico145 with TRPC5, which are the most important findings of this manuscript. However, the figure did not illustrate these detailed interactions very well. Protein residues involved in binding should be highlighted.

Reviewer #3 (Remarks to the Author):

This manuscript by Wright et al described cryo-EM structures of human TRPC5, with Pico145 and

ZnCl₂. TRPC5 belongs to TRP Canonical subfamily of TRP channel superfamily, which is a receptor-activated non-selective calcium-permeable cation channel. Several cryo-EM structures in the same family have been recently reported including TRPC3, TRPC4, TRPC5 and TRPC6. TRPC5 is a potential drug target for the treatment of kidney disease and CNS disorders. Structural insights of small molecular Pico 145 will help medicine chemist improve the affinity of new ligands and further promote the development of clinical trials. Taken together, this is an interesting work. However, there are several issues needed to be addressed to make this paper more solid and reliable.

Major points:

1. The main contribution of this manuscript is the novel binding site of Pico 145 in TRPC5, which is different from that of other TRP channels. The density of Pico 145 is not bad in the manuscript, but the original map should be provided for confirming density of small molecular since it is pretty common to observe lipid-like densities in cryo-EM maps of membrane proteins. In addition, three mutations including Q573T, F576A and W577A were generated for confirming Pico 145 binding site, but F576A mutation also result in an inactive channel in the apo structure of TRPC5 by Duan et al. (Science Advance, 2019).

2. The authors performed mutational study for the binding of small molecular using intracellular Ca²⁺ measurements. Is any patch clamp experiment supporting the Ca²⁺ measurements since Ca²⁺ measurement is not the gold standard methods for activity test and sometimes provide false positive data?

3. For the "The TRPC5:Pico145 structure reveals a putative, conserved zinc binding site" part, the authors claim that they find a conserved zinc binding site in TRPC5. They confirm a non-protein density as Zinc density according to the PDB motif query search, this density maybe due to the noise since the region in TRPC5 is very flexible. Is any functional data supporting this hypothesis? In the introduction part, the authors should mention more background and previous work about the Zinc binding site in TRPC subfamily.

Minor points:

1. The authors should label residues interacting with Zinc in Figure 5 and all related figures.

2. The authors should talk about the possible entry pathway of Pico 145 in the Discussion part.

Point-by-point response to editor's and reviewer's comments

We thank the editor and reviewers for their thorough evaluation of our study and their constructive comments, which have allowed us to improve the clarity and presentation of our work. We have made every effort to respond positively to all suggestions, and we have included new data to support our claims regarding the use of intracellular calcium recordings for the profiling of TRPC5 mutant channels. We have included point-by-point responses to all comments (including quoted text for any major changes) and provide a highlighted document to show all changes made to the manuscript.

Editor:

1. While all reviewers find this study interesting, they raised concerns regarding some discrepancy between text and data...

We have revised the text and data figures throughout, incorporating the reviewers' suggestions, to ensure clarity between text and data. A document with tracked changes is provided.

2. and a lack of functional data to support their claims for a conserved binding site of Pico145 between TRPC4 and TRPC5 channels...

The reviewer is correct that TRPC4 and TRPC5 are physiologically different, but our studies do not address physiology. Pico145 has near-identical pharmacological effects on TRPC5 and TRPC4, with the only difference being a shift in potency (J. Biol. Chem. 2017, 292, 8158). In addition, the xanthine/lipid binding site is almost identical in TRPC5 and TRPC4, in terms of shape of the pocket, the presence of a phospholipid in the apo structures (**Supplementary Figure 5B** vs **Supplementary Figure 5C**), and the local sequence (**Supplementary Table 1**). Moreover, the TRPC5 residues predicted to interact with Pico145 are highly conserved in TRPC4 (the only difference being that V579 in TRPC5 is an isoleucine in TRPC4; **Supplementary Table 1**).

These combined findings led to the hypothesis that Pico145 binds in the same site in both channels. Our docking simulations suggest that a molecular interaction between Pico145 and the TRPC4 lipid binding site is indeed possible: the minor differences in the TRPC4 binding site do not hinder binding of Pico145. We do see a small difference in lowest energy pose, which may be one factor contributing to higher potency of Pico145 against TRPC4. We have modified the paragraph on docking to TRPC4, and have been very conservative in our phrasing when discussing these docking results and their implications (see quoted text in response to reviewer 1 below).

3. ...and a conserved zinc-binding site in TRPC5.

The identification of a putative zinc binding site in TRPC5 was an observation that we felt would be of interest to the wider field because of recent reports about the role of zinc ions in TRPC channel modulation (Mol. Neurobiol. 2019, 56, 2822). Although apparently overlooked in previous analyses of TRPC channel structures, after our initial preprint of this manuscript, the observation of a zinc binding site was subsequently reported in a preprint by Song et al. as well (bioRxiv 2020, DOI 10.1101/2020.04.21.052910; we now refer to this preprint in our manuscript). Evaluation of the functional relevance of the putative zinc binding site would require substantial further experimentation and is beyond the scope of this paper, which focusses on the pharmacology of xanthines. We have made it clear within the manuscript that further characterisation of this site is required, but that the possibility of a zinc binding site is of great interest in understanding the regulation of TRPC channels. We have now adjusted the text in both the results and discussion sections (see quoted text in responses to reviewers below).

4. We also ask you to improve your data presentation, ...

We have improved the data presentation according to suggestions made by the reviewers. We are not certain what reviewer 2 meant with: "In addition, segmented cryo-EM density maps should be shown for assessment of local map quality." We hope the reviewer agrees that the new figures, explicitly stated contour levels, and the availability of density maps for review purposes, will allow full assessment of local map quality. We have

also ensured that the Pico145 density is displayed along with that of the surrounding residues to show that the inhibitor is not fitted within the “noise” of the map, but can be modelled unambiguously.

5. ...to provide the original density maps, ...

We have made the original density maps, as well as the PDB file, available for review purposes.

6. ...and to perform patch-clamp experiments.

We have patch-clamp capability and published extensive data comparing calcium and patch assays for evaluation of TRPC1/4/5 inhibitors and activators (for selected examples with (-)-englerin A, Pico145 and AM237, see Angew. Chem. Int. Ed. 2015, 54, 3787; J. Biol. Chem. 2017, 292, 8158 and Br. J. Pharmacol. 2019, 196, 3924). The data show suitability of our calcium assays and that little would be gained by performing patch-clamp as well. With some compounds there can be problems when using a fluorometric calcium assays, but this is not the case with the compounds studied here, as demonstrated in the aforementioned references. We have now also added new data in **Supplementary Figure 10** (shown in response to reviewer 3 below), which show that neither (-)-englerin A nor AM237 (the two TRPC5 activators used to profile our TRPC5 variants) gives non-specific calcium signals up to 5 μ M (the highest concentrations of EA and AM237 used in our mutant profiling). Although patch-clamp may provide useful biophysical information on the (mutant) channel properties, such information is not relevant to the aims of this study, which focusses on the analysis of molecular interactions of the xanthine-based TRPC1/4/5 inhibitor Pico145 with the lipid binding site of TRPC5.

Reviewer 1:

1. However, it would be better and more convincing for authors to use Pico145, but not the AM237 in testing the response of TRPC5 with mutations in putative binding site for two reasons: first the Pico145 and Am235 is not exactly same in terms of modulating the function of TRPC5, and secondly the Pico145 was used in this study to form the TRPC5-Pico145 complex.

We deliberately decided to use the TRPC5 activator AM237 instead of the TRPC5 inhibitor Pico145 to profile the effects of mutations in the lipid/xanthine binding site of TRPC5. The reason is that the only TRPC5 activators that give robust responses with transiently transfected TRPC5 variants, (-)-englerin A (EA) and AM237, are both competitively inhibited by Pico145. A potential reason is that all compounds bind to a similar site, a hypothesis (discussed in our manuscript) that is consistent with docking studies and the observation that mutations in the lipid/xanthine binding site affect the EC₅₀ of both EA and AM237 (**Figure 4; Table 2**).

Other activation mechanisms that are thought to involve intracellular signalling (S1P, carbachol/MIR) may affect changes to the nature of the phospholipid in the xanthine/lipid binding site, and could therefore interfere with the profiling of Pico145 as an inhibitor of mutant channels. It may be possible that an activator such as Gd³⁺ is compatible with profiling of Pico145 against TRPC5 mutants. However, such studies would require substantial experimentation, as well as the creation of stable cell lines to mitigate the low efficacy of Gd³⁺ as a TRPC5 agonist, and are not guaranteed to give meaningful data, especially because (as highlighted in our introduction) combinations of Gd³⁺ and Pico145 can give complex functional effects on TRPC1/4/5 channels (J. Biol. Chem. 2017, 292, 8158). Because of Covid-19, our laboratories are still closed and we do not know when we would have sufficient access again to conduct such experiments. We have further emphasised our reasons for using AM237 rather than Pico145 in the text as follows:

“The inhibitory potency of Pico145 on TRPC1/4/5 channels is dependent on the EA concentration used in assays,²⁴ which was expected to complicate analysis of Pico145 responses of TRPC5 variants. Other commonly used TRPC5:C5 activators such as S1P and carbachol (at least partially) act through intracellular signalling, and may interfere with composition of phospholipids bound to TRPC5, while Gd³⁺ has complex effects in the presence of Pico145.²⁴ Therefore, we decided to test TRPC5 responses to AM237, which is thought to occupy the same binding site as Pico145 (see above and ^{31,32}).”

2. Page3, line3-5 “Further negative stain analysis of the TRPC5 protein in the presence of Pico145 showed a similar monodispersity with no evidence for any gross conformational changes or disruption

of the tetramer.” The negative images of TRPC5 tetramer with/without Pico-145 (Supplementary Figure 2, panel B, C) do look different in terms of staining and appearance of particles. Please explain the difference.

In the original figures, the [+Pico145] image was of a region with higher particle density. We have now replaced it with one of the other images taken from the same grid during the same microscopy session (Supplementary Figure 2C). We used these negative stain EM data for quality control of protein samples, focussing on mono-dispersity of samples and lack of gross changes such as aggregation/loss of tetrameric structure. We have revised our text to emphasise this:

“Further negative stain analysis of the TRPC5 protein in the presence of Pico145 showed a similar monodispersity with no evidence for aggregation or disruption of the tetramer.”

3. Figure 1: please label the Pico-145 and putative zinc binding sites with clear text, something like “Pico145 binding site” or “Zinc binding site”.

The relevant sites have now been labelled in **Figure 1**.

4. Does the reference map used in the 3D reconstruction of TRPC5 without symmetry imposed come from the 3D reconstruction that has symmetry imposed?

Yes, the starting model was indeed symmetric. This strategy was based on our experience with other multimeric membrane proteins such as AcrB, where using a symmetrical starting model resulted in a non-symmetric reconstruction with two closed and one open site (*Microorganisms* **2020**, *8*, E943). This can be achieved by using a very low resolution starting model (e.g., 20-40 Å). Given the striking difference between the U-shaped density of a phospholipid and the density of Pico145, we would expect that, if differences in binding site occupancy had been present, they would have been resolved in our maps.

5. Page 4: “Comparison of our TRPC5 structure to the previously reported 2.9 Å mTRPC5 apo structure” should be “Comparison of our TRPC5 structure to the previously reported 2.8 Å mTRPC5 apo structure”

For the sake of clarity, we have now stated that the mTRPC5 structure reported by Duan *et al.* has a global resolution of 2.89 Å, which is consistent with their PDB file (6aei). We agree that the difference in resolution with our best structure (2.82 Å) is minimal, and therefore no longer state that our structure has the highest resolution reported so far (statement removed from page 10).

6. In figures 2, 3 (panel A), Supplementary figures 5, 6, and 10, the atomic models have the same colors as their corresponding EM density map, which gave the poor contrast and made them hard to see, this is especially bad for grey atomic models in grey density maps. Suggest to adjust the color to make the figures with better contrast.

The clarity of these figures has been improved by presenting the density maps and atomic models in different colours.

7. Page 5, First paragraph about the Figure 3a. In addition to W577, W573 and F576, there are many other amino acid side chains also involved in the interaction with Pico145, as described in the text (L521, Y524, C525, V579, L572, T603, G606, V610, V614), please also label these amino acids in the figure 3a to make them clear for general readers. This should also apply to Supplementary figure 7b (page 22).

These residues have now been labelled in the relevant figures.

8. Page 5, last paragraph: “During the optimisation of conditions for TRPC5:Pico145 cryo-grid preparation, we varied concentrations of TRPC5 and Pico145. As part of this process, an additional structure of TRPC5 in the presence of Pico145 was solved with TRPC5 at 1 mg·ml⁻¹ and Pico145 at

50 μM (compared to 2 $\text{mg}\cdot\text{ml}^{-1}$ TRPC5 and 100 μM Pico145 in the structure described above) to a global resolution of 2.9 Å. The density around the Pico145 binding site in this structure showed features consistent with both the bound phospholipid and Pico145, suggesting partial occupancy under these conditions (Supplementary Figure 6A-C). These data further support the observation that Pico145 can displace each of the four phospholipids bound to a tetrameric TRPC5 channel.”

The sample 2 $\text{mg}\cdot\text{ml}^{-1}$ TRPC5 with 100 μM Pico145 and the sample of 1 $\text{mg}\cdot\text{ml}^{-1}$ TRPC5 with 50 μM Pico145 should have same molar ratio of TRPC5-Pico145, but the 2.9 Å cryoEM map from later suggested the partial occupancy of Pico145. Please explain.

The reviewer correctly notes that the ratios in both samples are identical, and that, given the high potency of Pico145, a similar occupancy would intuitively be expected. We have no full explanation for our observation, but hypothesise that the difference in occupancy between the structures is the result of the sample preparation method: Pico145 was added to TRPC5 after purification and exchange into amphipol (see Methods), which may alter binding kinetics from those in the cellular context. Further studies would be required to find a definite answer, but these are beyond the scope and budget of our current investigation. For clarity, we have now amended the paragraph as follows:

“The density around the Pico145 binding site in this structure showed features consistent with both the bound phospholipid and Pico145, suggesting partial occupancy under these conditions (Supplementary Figure 6A-C). This observation may seem counter-intuitive, given the high potency of Pico145 and the fact that TRPC5:Pico145 ratios were identical in both samples. A reason could be that binding kinetics of Pico145 to purified TRPC5:C5 in amphipol may differ from those in the cellular context.”

9. Page 8, figure 4 and Page 23, supplementary figure 8, please label the amino acids that are involved in the interactions.

These residues have now been labelled.

10. Page 6 : “To gain insight into the potent inhibition of TRPC4 channels by Pico14523, we next docked Pico145 into the published mTRPC4 apo structure (PDB 5z96; the phospholipid was removed before docking). The docking suggests that Pico145 adopts a similar pose in TRPC4 and TRPC5, making similar interactions with conserved residues, particularly residues equivalent to the TRPC5 residues Q573, F576 and W577 (Supplementary Figure 9). These data suggest that the binding site of Pico145 is conserved between TRPC4 and TRPC5 channels.” Because the published mTRPC5 and mTRPC4 structure are almost identical, especially in the region where phospholipid binds, but their physiological functions do differ. Therefore, by just docking the Pico145 into the mTRPC4 structure without other supportive data, it’s pretty risky to draw such a conclusion.

The reviewer is correct that TRPC4 and TRPC5 are physiologically different, but our studies do not address physiology. Pico145 has near-identical pharmacological effects on TRPC5 and TRPC4, with the only difference being a shift in potency (*J. Biol. Chem.* **2017**, *292*, 8158). In addition, the xanthine/lipid binding site is almost identical in TRPC5 and TRPC4, in terms of shape of the pocket, the presence of a phospholipid in the apo structures (Supplementary Figure 5B vs Supplementary Figure 5C), and the local sequence (Supplementary Table 1). Moreover, the TRPC5 residues predicted to interact with Pico145 are highly conserved in TRPC4 (the only difference being that V579 in TRPC5 is an isoleucine in TRPC4; Supplementary Table 1).

These combined findings led to the hypothesis that Pico145 binds in the same site in both channels. Our docking simulations suggest that a molecular interaction between Pico145 and the TRPC4 lipid binding site is indeed possible: the minor differences in the TRPC4 binding site do not hinder binding of Pico145. We do see a small difference in lowest energy pose, which may be one factor contributing to higher potency of Pico145 against TRPC4. We have now carefully modified the text of this paragraph:

“Pico145 is also a potent inhibitor of other TRPC1/4/5 channels,^{24,25} including TRPC4:C4, for which a structure is available, and which has a lipid binding site that is near-identical to that of TRPC5:C5 in terms of shape, bound phospholipid and residues lining the pocket (Supplementary Figure 5B vs Supplementary

Figure 5C). Importantly, the TRPC5 residues predicted to interact with Pico145 are highly conserved in TRPC4, with the only difference being that V579 in TRPC5 is an isoleucine in TRPC4 (**Supplementary Table 1**). In order to test if Pico145 would be able to bind in the lipid binding site of TRPC4:C4, we next docked Pico145 into the published mTRPC4 apo structure (PDB 5z96; the phospholipid was removed before docking). The docking suggests that Pico145 adopts a similar pose in TRPC4 and TRPC5, making similar interactions with conserved residues, particularly residues equivalent to the TRPC5 residues Q573, F576 and W577 (**Supplementary Figure 9**). These data suggest that the binding site of Pico145 is conserved between TRPC4 and TRPC5 channels.”

11. Page 8: authors claim their TRPC5:Pico145 structure reveals a putative, conserved zinc binding site. If there is any EM density corresponding to this bound Zinc? Authors used sequence alignment of hTRPC5 and hURF1 to support their point, but at resolution of 2.8 Å with the presence of ZnCl₂, there should be an EM density showing the Zinc is indeed exist.

The local resolution in the putative zinc binding site is lower than the global resolution of 2.8 Å and therefore the local map quality is reduced. This could be because of flexibility and/or because of partial occupancy of the Zn²⁺ ion. The local resolution was insufficient to assess occupancy when C1 symmetry was applied. Therefore, we cannot unambiguously identify the density of a Zn²⁺ ion. We have added the following text to this results section:

“In the zinc binding domain, the densities were similar (and partially disordered) in the TRPC5:Zn²⁺ and TRPC5:Pico145 structures. Because of this disorder, and the lower local resolution, we could not unambiguously identify the density of the Zn²⁺ ion in either structure. These observations may suggest that this region is flexible and/or that fewer than four of the zinc binding sites are occupied at any one time, even in the presence of added ZnCl₂.”

Because the region is conserved in TRPC channels (**Supplementary Figure 11**) and the residues are consistent with a metal (most likely zinc) binding site, we felt that the identification of this putative (conserved) zinc binding site in TRPC5 would be of interest to the wider field because of recent reports about the role of zinc ions in TRPC channel modulation (*Mol. Neurobiol.* **2019**, *56*, 2822). Although apparently overlooked in previous analyses of TRPC channel structures, after our initial preprint of this manuscript, the observation of a zinc binding site was subsequently reported in a preprint by Song *et al.* as well (*bioRxiv* **2020**, DOI [10.1101/2020.04.21.052910](https://doi.org/10.1101/2020.04.21.052910); we now refer to this preprint in our manuscript).

12. Page 10, line 5: “the highest resolution TRPC5 structure to date.” This is not true, because ref. 34 “Cryo-EM structure of TRPC5 at 2.8 Å resolution reveals unique and conserved structural elements essential for channel function” by Duan et al reported a 2.8 Å TRPC5 structure already.

As explained above, we have removed this statement.

Reviewer 2:

1. Structural illustrations throughout the manuscript need to be improved for clarity. For instance, all structural illustrations with cryo-EM density maps were not of high quality. The contour levels were not specified. Structural elements and important residues mentioned in the text were not labeled in the figures. In addition, segmented cryo-EM density maps should be shown for assessment of local map quality.

We have improved the data presentation according to these suggestions. We are not certain what the reviewer means with: “In addition, segmented cryo-EM density maps should be shown for assessment of local map quality.” We hope the reviewer agrees that the new figures, explicitly stated contour levels, and the availability of density maps for review purposes, will allow full assessment of local map quality.

2. The authors stated that “We tentatively modelled residues 734 to 759, yet the position of these residues should be interpreted with caution”. Do the authors mean that this region was not confidently built? Maybe consider poly-Ala model for this region. Again, EM density should be shown.

As the density did not allow us to confidently build this region, we have removed it from the revised model. As this region is not near the different binding sites identified in this study, the absence of the region in the model does not affect our conclusions. We have opted for removal rather than replacement with a poly-Ala model to maximise the ease of using the PDB file for docking studies. We have updated the PDB file and the refinement statistics table (now **Table 1**) accordingly.

3. The authors stated that “This region showed that histidine 172 and cysteines 176, 178 and 181 all point towards a central density, consistent with metal ion binding (Figure 1B and Figure 5; see below for details)”. Cryo-EM density for Zn²⁺ was not shown.

The local resolution in the putative zinc binding site is lower than the global resolution of 2.8 Å and therefore the local map quality is reduced. This could be because of flexibility and/or because of partial occupancy of the Zn²⁺ ion. The local resolution was insufficient to assess occupancy when C1 symmetry was applied. Therefore, we cannot unambiguously identify the density of the Zn²⁺ ion. We have added the following text to this results section:

“In the zinc binding domain, the densities were similar (and partially disordered) in the TRPC5:Zn²⁺ and TRPC5:Pico145 structures. Because of this disorder, and the lower local resolution, we could not unambiguously identify the density of the Zn²⁺ ion in either structure. These observations may suggest that this region is flexible and/or that fewer than four of the zinc binding sites are occupied at any one time, even in the presence of added ZnCl₂.”

4. Fig. 3 is supposed to show detailed interactions of Pico145 with TRPC5, which are the most important findings of this manuscript. However, the figure did not illustrate these detailed interactions very well. Protein residues involved in binding should be highlighted.

The relevant residues have now been highlighted in **Figure 3**.

Reviewer 3:

1. The main contribution of this manuscript is the novel binding site of Pico 145 in TRPC5, which is different from that of other TRP channels. The density of Pico 145 is not bad in the manuscript, but the original map should be provided for confirming density of small molecular since it is pretty common to observe lipid-like densities in cryo-EM maps of membrane proteins.

We have made the original density maps, as well as the PDB, available for review purposes. We have also explicitly stated the contour levels in the legends of all figures with density maps.

2. In addition, three mutations including Q573T, F576A and W577A were generated for confirming Pico 145 binding site, but F576A mutation also result in an inactive channel in the apo structure of TRPC5 by Duan et al. (Science Advance, 2019).

Duan *et al.* described a double mutant (F576A,W577A), which did not respond to 100 nM (-)-englerin A (Figure 6E in *Sci. Adv.* **2019**, *5*, eaaw7935). No data were presented for the single mutants F576A and/or W577A. In addition, Duan *et al.* did not report the use of (-)-englerin A at concentrations higher than 100 nM. Our data demonstrate that each of the two mutations can increase the EC₅₀ of (-)-englerin A (ca. 275-fold for F476A and ca. 40-fold for W577A). This may explain why Duan *et al.* found that the double mutant did not respond to 100 nM (-)-englerin A. We have added the following sentence to our manuscript:

*“In addition, the large decreases in EA potency against F576A (ca. 275-fold) and W577A (ca. 40-fold) (Table 2) may explain why Duan et al. found that the mTRPC5 variant (F576A,W577A) was unresponsive to 100 nM EA.”*³⁵

3. The authors performed mutational study for the binding of small molecular using intracellular Ca²⁺ measurements. Is any patch clamp experiment supporting the Ca²⁺ measurements since Ca²⁺ measurement is not the gold standard methods for activity test and sometimes provide false positive data?

We have patch-clamp capability and published extensive data comparing calcium and patch assays for evaluation of TRPC1/4/5 inhibitors and activators (for selected examples with (-)-englerin A, Pico145 and AM237, see Angew. Chem. Int. Ed. 2015, 54, 3787; J. Biol. Chem. 2017, 292, 8158 and Br. J. Pharmacol. 2019, 196, 3924). The data show suitability of our calcium assays and that little would be gained by performing patch-clamp as well. With some compounds there can be problems when using a fluorometric calcium assays, but this is not the case with the compounds studied here, as demonstrated in the aforementioned references. We have now also added new data in **Supplementary Figure 10** (see below), which show that neither (-)-englerin A nor AM237 (the two TRPC5 activators used to profile our TRPC5 variants) gives non-specific calcium signals up to 5 μM (the highest concentrations of EA and AM237 used in our mutant profiling). Although patch-clamp may provide useful biophysical information on the (mutant) channel properties, such information is not relevant to the aims of this study, which focusses on the analysis of molecular interactions of the xanthine-based TRPC1/4/5 inhibitor Pico145 with the lipid binding site of TRPC5. We have added the following text:

“Note that in wild-type HEK 293 cells, application of up to 5 μM of EA or AM237 did not result in increases in $[\text{Ca}^{2+}]_i$ (Supplementary Figure 10), demonstrating that the signals detected in Figure 4 are the result of TRPC5 channel activation rather than other calcium handling processes.”

Wild-type HEK 293 cells

Supplementary Figure 10: The TRPC5 activators EA and AM237 do not cause non-specific calcium signals in wild-type HEK 293 cells. A) $[\text{Ca}^{2+}]_i$ measurements from a single 96-well plate (N = 1) showing that 0.3-5000 nM EA has no effect on $[\text{Ca}^{2+}]_i$ in wild-type HEK 293 cells. B) $[\text{Ca}^{2+}]_i$ measurements from a single 96-well plate (N = 1) showing that 0.3-5000 nM AM237 has no effect on $[\text{Ca}^{2+}]_i$ in wild-type HEK 293 cells.

4. For the “The TRPC5:Pico145 structure reveals a putative, conserved zinc binding site” part, the authors claim that they find a conserved zinc binding site in TRPC5. They confirm a non-protein density as Zinc density according to the PDB motif query search, this density maybe due to the noise since the region in TRPC5 is very flexible. Is any functional data supporting this hypothesis? In the introduction part, the authors should mention more background and previous work about the Zinc binding site in TRPC subfamily.

The local resolution in the putative zinc binding site is lower than the global resolution of 2.8 Å and therefore the local map quality is reduced. This could be because of flexibility and/or because of partial occupancy of the Zn²⁺ ion. The local resolution was insufficient to assess occupancy when C1 symmetry was applied, and we agree that the density could be noise. Therefore, we cannot unambiguously identify the density of a Zn²⁺ ion. We have added the following text to this results section:

“In the zinc binding domain, the densities were similar (and partially disordered) in the TRPC5:Zn²⁺ and TRPC5:Pico145 structures. Because of this disorder, and the lower local resolution, we could not unambiguously identify the density of the Zn²⁺ ion in either structure. These observations may suggest that this region is flexible and/or that fewer than four of the zinc binding sites are occupied at any one time, even in the presence of added ZnCl₂.”

Because the region is conserved in TRPC channels (**Supplementary Figure 11**) and the residues are consistent with a metal (most likely zinc) binding site, we felt that the identification of this putative (conserved) zinc binding site in TRPC5 would be of interest to the wider field because of recent reports about the role of zinc ions in TRPC channel modulation (Mol. Neurobiol. 2019, 56, 2822; now highlighted in our introduction, as suggested by the reviewer). Although apparently overlooked in previous analyses of TRPC channel structures, after our initial preprint of this manuscript, the observation of a zinc binding site was subsequently reported in a preprint by Song *et al.* as well (bioRxiv 2020, DOI 10.1101/2020.04.21.052910; we now refer to this preprint in our manuscript). Importantly, this metal binding site is different from the calcium/sodium binding site identified by Duan *et al.* (Sci. Adv. 2019, 5, eaaw7935). Evaluation of the functional relevance of the putative zinc binding site would require substantial further experimentation (which is currently not possible because of Covid-19 related lab closure) and is beyond the scope of this paper, which focusses on the pharmacology of xanthines. We have made it clear within the manuscript that further characterisation of this site is required, but that the possibility of a zinc binding site is of great interest in understanding the regulation of TRPC channels. We have amended the penultimate discussion paragraph as follows:

“Intracellular Zn²⁺-dependent activation of TRPC5 channels was recently reported to contribute to oxidative neuronal death,²³ but the molecular mechanism of Zn²⁺ regulation of TRPC5 was not determined. We have identified a putative intracellular zinc binding site of TRPC5, which is conserved in all TRPC channels. Cryo-EM of TRPC5 in the presence of ZnCl₂ resulted in a further TRPC structure determined to a resolution of 2.8 Å. Presence of ZnCl₂ did not change the overall structure compared to TRPC5:Pico145, and the density of the Zn²⁺ ion could not be unambiguously identified in either structure. These observations suggest flexibility of the region or partial maximum occupancy of the zinc binding site, and may indicate that zinc is already bound in TRPC5:Pico145, or that the role of zinc in TRPC5 modulation is subtle. A subtle role of Zn²⁺ in TRPC5 modulation would also be consistent with the delayed Zn²⁺-mediated [Ca²⁺]_i increase observed in calcium imaging experiments, and the small TRPC5 currents evoked by intracellular application of ZnCl₂.²³ Part of the putative zinc binding site has also been implicated in TRPC5 glutathionylation: mutation of C176, C178 or C181 prevented TRPC5 opening in response to glutathionylation.⁴¹ A separate report proposed that palmitoylation of C181 is required for correct trafficking of TRPC5 to the plasma membrane in striatal neurons.⁴² These data suggest that this small domain may be an important regulatory node of TRPC channels, and that further experiments are needed to fully understand its role in TRPC channel biology.”

5. The authors should label residues interacting with Zinc in Figure 5 and all related figures.

The relevant residues have now been labelled in Figure 5 and all other relevant figures.

6. The authors should talk about the possible entry pathway of Pico 145 in the Discussion part.

We have added the following sentences to the first paragraph of the Discussion section:

“We hypothesise that Pico145 initially integrates with the lipid membrane, before entering the lipid binding site between two TRPC5 subunits and replacing the bound phospholipid. This hypothesis is consistent with the hydrophobic nature of Pico145, and with the observation that the effect of Pico145 is not as rapid as would be expected for a pore blocker.²⁴”

REVIEWERS' COMMENTS:

Reviewer #1 (Remarks to the Author):

Overall:

The authors of the "Cryo-EM structures of human TRPC5 reveal interaction of a xanthine-based TRPC1/4/5 inhibitor with a conserved lipid binding site" by David J. Wright et al, have addressed most of the questions/concerns I had previously to my satisfaction, make the manuscript much better.

Following are some minor suggestions/comments for authors:

1. In the rebuttal letter, "Point-by-point response to editor's and reviewer's comments", section for Reviewer 1:

"5. Page4: "Comparison of our TRPC5 structure to the previously reported 2.9 Å mTRPC5 apo structure" should be "Comparison of our TRPC5 structure to the previously reported 2.8 Å mTRPC5 apo structure" "

"For the sake of clarity, we have now stated that the mTRPC5 structure reported by Duan et al. has a global resolution of 2.89 Å, which is consistent with their PDB file (6aei). We agree that the difference in resolution with our best structure (2.82 Å) is minimal, and therefore no longer state that our structure has the highest resolution reported so far (statement removed from page 10). " It seems that authors didn't really read the original paper by Duan et al. and still said in there manuscript "Comparison of our TRPC5 structure to the previously reported 2.89 Å mTRPC5 apo structure (PDB 6aei)³⁵ revealed....." (Results, Structure of a human TRPC5:C5 channel in complex with Pico145, first line in the last paragraph).

Duan et al.³⁵ clearly stated the "2.8 Å resolution" throughout their paper, and in the FSC curve showed in Fig. S2, panel B the 2.81 Å at 0.143 FSC.

2. It's better to use "cryoEM density map" rather than "electron density map" (1> Figure 3 legend, A) the electron density map determined by 2> Results, The TRPC5:Pico145 structure reveals a putative, conserved zinc binding site, reciprocal 4th line:present in the electron density maps of all TRPC channel structures.....)

Not like the case in the x-ray crystallography, the interaction between electrons and individual atoms are different and it's not scientifically accurate to describe the cryoEM map as the "electron density map".

3. The Figure 3, panel A is inconsistent with the legend and the text, please modify the figure accordingly (color of monomer 1 and 2, and the label of Q573, F576 and W577)

4. Materials and Methods, Negative stain electron microscopy:

Line 3: After charging, should be "After glow-discharging,"

5. Materials and Methods, Image processing:

Please cite the relevant references for software used, such as MotionCor2, Gctf, crYOLO, et al.

6. Materials and Methods, Creation of PDB file:

Please cite the relevant references for software used, such as Phenix, WinCoot, UCSF Chimera et al.

Reviewer #2 (Remarks to the Author):

The authors have made satisfactory revisions.

Reviewer #3 (Remarks to the Author):

I am satisfied with the response to my previous concerns.No further comments.

Point-by-point response to editor's and reviewer's comments – round 2

We thank the editor and reviewers for their thorough evaluation of our revised study, their constructive comments, and their attention to detail. We have now addressed the minor comments by the editor and reviewer 1 (please see below) and provide a highlighted document to show all changes made to the manuscript. Additional editorial requests have been addressed as detailed in the separate, annotated AIP table.

Editor:

1. We ask you to please address the minor comments of reviewer #1 in the revised manuscript and to cite a preprint as a non-peer-reviewed study.

We have addressed the minor comments by reviewer #1 and have now acknowledged the preprint by Song et al. as a non-peer-reviewed study in the final paragraph of our discussion, as follows:

“To the best of our knowledge, our data provide the first structural insight into TRPC1/4/5 channel modulation and suggest direct modulatory roles for (phospho)lipids and Zn²⁺ ions. Our data are consistent with the description of a structure of hTRPC5 in complex with the xanthine HC-070 in a non-peer-reviewed study by Song et al.,⁴³ which appeared shortly after the initial disclosure of our study on bioRxiv. These studies lay the foundations for the structure-based design of TRPC1/4/5 modulators, and may therefore support the development of new TRPC1/4/5 chemical probes and drug candidates for an increasing number of therapeutic areas.”

The reference to our own preprint (Bauer et al. 2020) has now been updated as this paper has now been published.

Reviewer 1:

1. In the rebuttal letter, “Point-by-point response to editor's and reviewer's comments”, section for Reviewer 1: “5. Page4: “Comparison of our TRPC5 structure to the previously reported 2.9 Å mTRPC5 apo structure” should be “Comparison of our TRPC5 structure to the previously reported 2.8 Å mTRPC5 apo structure” ”

“For the sake of clarity, we have now stated that the mTRPC5 structure reported by Duan et al. has a global resolution of 2.89 Å, which is consistent with their PDB file (6aei). We agree that the difference in resolution with our best structure (2.82 Å) is minimal, and therefore no longer state that our structure has the highest resolution reported so far (statement removed from page 10). ”

It seems that authors didn't really read the original paper by Duan et al. and still said in there manuscript “Comparison of our TRPC5 structure to the previously reported 2.89 Å mTRPC5 apo structure (PDB 6aei)³⁵ revealed.....” (Results, Structure of a human TRPC5:C5 channel in complex with Pico145, first line in the last paragraph). Duan et al.³⁵ clearly stated the “2.8 Å resolution” throughout their paper, and in the FSC curve showed in Fig. S2, panel B the 2.81 Å at 0.143 FSC.

We thank the reviewer for pointing this out. We had not inspected the FSC curve in Figure S2B of the Supplementary Materials of this paper, and had instead used the reported resolution in the corresponding PDB 6aei (which we used for comparison to our structures). We assume that the authors used the 2.89 Å structure before CTF & beam-tilt refinement (green in Figure S2B) for the model deposited in the PDB rather than the 2.81 Å structure (blue in Figure S2B).

Consistent with the structure reported in the paper and as requested by the reviewer, we now refer to the ‘...previously reported 2.8 Å mTRPC5 apo structure.’ by Duan et al. instead.

2. It's better to use “cryoEM density map” rather than “electron density map” (1> Figure 3 legend, A) the electron density map determined by 2> Results, The TRPC5:Pico145 structure reveals a putative, conserved zinc binding site, reciprocal 4th line:present in the electron density maps of all TRPC channel structures.....)

Not like the case in the x-ray crystallography, the interaction between electrons and individual atoms are different and it's not scientifically accurate to describe the cryoEM map as the “electron density map”.

We have now changed this according to the suggestions.

3. The Figure 3, panel A is inconsistent with the legend and the text, please modify the figure accordingly (color of monomer 1 and 2, and the label of Q573, F576 and W577)

We have now corrected this Figure 3 and the corresponding figure legend.

4. Materials and Methods, Negative stain electron microscopy: Line 3: After charging, should be “After glow-discharging,”

We have now corrected this as suggested.

5. Materials and Methods, Image processing: Please cite the relevant references for software used, such as MotionCor2, Gctf, crYOLO, et al.

We have now included the following references:

48. Zheng, S. Q. *et al.* MotionCor2: anisotropic correction of beam-induced motion for improved cryo-electron microscopy. *Nat. Methods* **14**, 331–332 (2017)
49. Zhang, K. Gctf: Real-time CTF determination and correction. *J. Struct. Biol.* **193**, 1–12 (2016).
50. Wagner, T. *et al.* SPHIRE-crYOLO is a fast and accurate fully automated particle picker for cryo-EM. *Commun. Biol.* **2**, 1–13 (2019).
51. Rohou, A. & Grigorieff, N. CTFFIND4: Fast and accurate defocus estimation from electron micrographs. *J. Struct. Biol.* **192**, 216–221 (2015)
6. Materials and Methods, Creation of PDB file: Please cite the relevant references for software used, such as Phenix, WinCoot, UCSF Chimera et al.

We have now included the following references:

52. Adams, P. D. *et al.* PHENIX : a comprehensive Python-based system for macromolecular structure solution. *Acta Crystallogr. Sect. D Biol. Crystallogr.* **66**, 213–221 (2010).
53. Emsley, P., Lohkamp, B., Scott, W. G. & Cowtan, K. Features and development of Coot. *Acta Crystallogr. Sect. D Biol. Crystallogr.* **66**, 486–501 (2010).
54. Pettersen, E. F. *et al.* UCSF Chimera—A visualization system for exploratory research and analysis. *J. Comput. Chem.* **25**, 1605–1612 (2004).